# Causal Discovery in Probabilistic Networks with an Identifiable Causal Effect

## Abstract

Causal identification is at the core of the causal inference literature, where complete algorithms have been proposed to identify causal queries of interest. The validity of these algorithms hinges on the restrictive assumption of having access to a correctly specified causal structure. In this work, we study the setting where a probabilistic model of the causal structure is available. Specifically, the edges in a causal graph are assigned probabilities which may, for example, represent degree of belief from domain experts. Alternatively, the uncertainly about an edge may reflect the confidence of a particular statistical test. The question that naturally arises in this setting is: Given such a probabilistic graph and a specific causal effect of interest, what is the subgraph which has the highest plausibility and for which the causal effect is identifiable? We show that answering this question reduces to solving an NP-hard combinatorial optimization problem which we call the edge ID problem. We propose efficient algorithms to approximate this problem, and evaluate our proposed algorithms against real-world networks and randomly generated graphs.

## 1 Introduction

A large proportion of questions of interest in various fields including but not limited to psychology, social sciences, behavioural sciences, medical research, epidemiology, economy, etc. are causal in nature [21, 13, 2]. In order to estimate causal effects, the gold standard is performing controlled interventions and experiments. Unfortunately, such experiments can be prohibitively expensive, unethical, or impractical (consider, for example, an experiment in which participants are required to smoke in order to understand the links to cancer) [3, 5]. In contrast, non-experimental data are comparatively abundant, and no expensive interventions are required to generate such data. This has motivated the development of numerous techniques for understanding whether a causal query can be answered using observational data. Specifically, if a particular causal query is *identifiable*, it means it can be expressed as a function of the observational distribution, and thus estimated from observational data (at least in principle).

A significant body of the causal inference literature is dedicated to the identification problem [18, 13, 16, 7, 12]. In particular, Huang and Valtora presented a complete algorithmic approach to decide the identifiability of a specific query, and proved that Pearl's do calculus is complete, in the sense that if a causal query is identifiable, a sequence of do calculus rules can be applied to derive an identification expression for that query [6]. Furthermore, Shpitser and Pearl provided a graphical criteria to decide the identifiability, based on the *hedge* criterion [16]. However, all of these results hinge on full specification of the causal structure, i.e., access to a correctly specified Acyclic Directed Mixed Graph (ADMG) that models the causal dynamics of the system. This requirement is restrictive in a number of ways. Firstly, the causal identification problem is concerned with inference from the observational data, but the ADMG cannot be inferred from the observational distribution alone.

Submitted to 36th Conference on Neural Information Processing Systems (NeurIPS 2022). Do not distribute.

Secondly, structure learning methods rely heavily on statistical tests, which are prone to errors arising from lack of sufficient data and method-specific limitations [15] which can result in misspecification of the causal structure.

As opposed to full specification of the causal structure, we propose the setting in which we only have access to a probabilistic model of the causal structure. For instance, an ADMG $\mathcal{G}$ is given along with probabilities assigned to each edge of $\mathcal{G}$. An example is shown in Figure 1a. These probabilities could represent uncertainties arising from statistical tests, or the strength of belief of domain experts concerning the plausibility of the existence of an edge. Under this setting, each ADMG on the set of vertices of $\mathcal{G}$ is assigned its own plausibility score. Since the causal structure is not deterministic anymore, answering questions such as "*is the causal effect $P(Y|do(X))$ identifiable?*" also becomes probabilistic in nature. One can compare the overall plausibility of different subgraphs in which the causal effect is identifiable, and then select the graph which maximises the plausibility. Indeed, identification is often assumed on the basis of ignorability (i.e., no unobserved confounders exist) [8, 14], thus the use of probabilistic models enables us to quantify the strength of such an assumption.

In this work, for a specific causal query $P(Y|do(X))$, we first answer the question "which graph has the highest plausibility among those compliant with the probabilistic ADMG model that renders $P(Y|do(X))$ identifiable?" The answer to this question then shows us with what confidence we can carry out the causal identification task using the combination of the data at hand and the corresponding probabilistic model.

Noting that the causal identification task is carried out through an identification formula which is based on the causal structure, our second focus is on deriving an identification formula for a given causal query that holds with the highest probability. This problem is different from the former in the sense that a single identification formula can be valid with respect to a set of different graphs. Therefore, the probability that a given identification formula is valid for a causal query would be the aggregate probability of all graphs on which this formula is valid. We shall illustrate this point in more detail through Example 1 in Section 2. To identify the most probable identification formula, we first show that if an identification formula is valid w.r.t. a causal graph, it is also valid w.r.t. all its edge-induced subgraphs. Afterwards, we propose a surrogate problem (see Problem 2 in Section 2.1) that recovers a causal graph with highest aggregated probability of its subgraphs. Both problems discussed in this work are aimed at evaluating the plausibility of performing causal identification for a specific query given a dataset and a non-deterministic model describing the causal structure.

To sum up, our main contributions are as follows.

1. We study the problem of causal identifiability in probabilistic causal models, where there are uncertainties about the existence of edges and whether a given causal effect is identifiable. More precisely, we consider two problems: 1) finding the most probable graph that renders a desired causal query identifiable, and 2) finding the graph with the highest aggregate probability over its edge-induced subgraphs that renders a desired causal query identifiable.

2. We show that both aforementioned problems reduce to a special combinatorial optimization problem which we call the *edge ID problem*. We prove that the edge ID problem is NP-hard, and thus, so are both of the problems we discussed.

3. We propose several exact and heuristic algorithms for the aforementioned problems.

In Section 2, we introduce the terminology and formally define the two problems we are considering in this work. In Section 3, we show that both of these problems are equivalent to the edge ID problem. Furthermore, we show that the edge ID problem is NP-hard. We discuss algorithmic approaches (both exact and heuristic) in Section 4. Empirical evaluations of our algorithms are presented in Section 5. Proofs and accompanying code are provided in the appendices and in supplementary material, respectively.

## 2 Preliminaries

We utilize small letters for variables, and capital letters for sets of variables. Calligraphic letters are used to denote graphs. An acyclic directed mixed graph (ADMG) $\mathcal{G} = (V^{\mathcal{G}}, E_d^{\mathcal{G}}, E_b^{\mathcal{G}})$ is defined as an acyclic graph on the vertices $V^{\mathcal{G}}$, where $E_d^{\mathcal{G}} \subseteq V^{\mathcal{G}} \times V^{\mathcal{G}}$ and $E_b^{\mathcal{G}} \subseteq \binom{V^{\mathcal{G}}}{2}$ are the set of directed and bidirected edges among the vertices, respectively. With slight abuse of notation, if $e \in E_d^{\mathcal{G}} \cup E_b^{\mathcal{G}}$, we

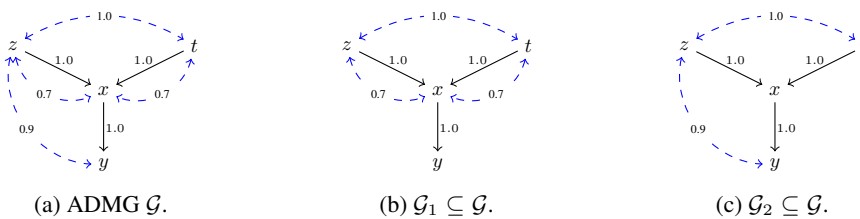

(a) ADMG $\mathcal{G}$.  (b) $\mathcal{G}_1 \subseteq \mathcal{G}$.  (c) $\mathcal{G}_2 \subseteq \mathcal{G}$.

Figure 1: (a) An example of a probabilistic ADMG $\mathcal{G}$ with corresponding edge probabilities. (b) and (c) are two different subgraphs of $\mathcal{G}$ in which $Q[y]$ is identifiable.

90  write $e \in \mathcal{G}$. We use $\mathcal{G}' \subseteq \mathcal{G}$ when $\mathcal{G}'$ is an edge-induced subgraph of $\mathcal{G}$, i.e., $\mathcal{G}' = (V^{\mathcal{G}'}, E_d^{\mathcal{G}'}, E_b^{\mathcal{G}'})$,
91  where $V^{\mathcal{G}'} = V^{\mathcal{G}}$ and $E_i^{\mathcal{G}'} \subseteq E_i^{\mathcal{G}}$ for $i \in \{b, d\}$. We denote by $\mathcal{G}[X]$ the vertex-induced subgraph
92  of $\mathcal{G}$ over the subset of vertices $X \subseteq V^{\mathcal{G}}$. For a set of vertices $X$, we denote by $Anc_{\mathcal{G}}(X)$ the set of
93  vertices in $\mathcal{G}$ that have a directed path to $X$. Note that $X \subseteq Anc_{\mathcal{G}}(X)$. Let $P_X(Y)$ be a shorthand for
94  $P(Y|do(X))$, and $P^M(\cdot)$ denote the distribution of variables described by the causal model $M$.

**Definition 1** (Identifiability [13]). *Given a causal ADMG $\mathcal{G} = (V^{\mathcal{G}}, E_d^{\mathcal{G}}, E_b^{\mathcal{G}})$, and two disjoint*
96  *subsets of variables $X, Y \subseteq V^{\mathcal{G}}$, the causal effect of $X$ on $Y$, denoted by $P_X(Y)$, is identifiable in $\mathcal{G}$ if*
97  $P_X^{M_1}(Y) = P_X^{M_2}(Y)$ *for any two models $M_1$ and $M_2$ that induce $\mathcal{G}$ and $P^{M_1}(V^{\mathcal{G}}) = P^{M_2}(V^{\mathcal{G}}) > 0$.*

**Definition 2** (Valid identification formula). *For a causal ADMG $\mathcal{G}$ over variables $V^{\mathcal{G}}$ and a causal*
99  *query $P_X(Y)$, we say a functional $\mathcal{F}$ defined on the probability space over $V^{\mathcal{G}}$ is a valid identification*
100  *formula for $P_X(Y)$ in $\mathcal{G}$ if $P_X^{M_1}(Y) = P_X^{M_2}(Y) = \mathcal{F}(P^{M_1}(V^{\mathcal{G}})) = \mathcal{F}(P^{M_2}(V^{\mathcal{G}}))$ for any two*
101  *models $M_1$ and $M_2$ that induce $\mathcal{G}$ and $P^{M_1}(V^{\mathcal{G}}) = P^{M_2}(V^{\mathcal{G}}) > 0$.*

102  For any query $P_X(Y)$, let $[\mathcal{G}]_{Id(P_X(Y))}$ denote the set of subgraphs of $\mathcal{G}$ in which $P_X(Y)$ is iden-
103  tifiable (note that if $\mathcal{G}$ is complete graph, $[\mathcal{G}]_{Id(P_X(Y))}$ is the set of all graphs in which $P_X(Y)$ is
104  identifiable.) We denote by $Q[Y]$ the causal effect of $V \setminus Y$ on $Y$, i.e., $Q[Y] = P(Y|do(V \setminus Y))$.

**Definition 3** (District [4]). *For ADMG $\mathcal{G} = (V^{\mathcal{G}}, E_d^{\mathcal{G}}, E_b^{\mathcal{G}})$, let $\mathcal{G}_{\leftrightarrow}$ denote the edge-induced subgraph*
106  *of $\mathcal{G}$ over its bidirected edges. $X \subseteq V^{\mathcal{G}}$ is a district (aka c-component) in $\mathcal{G}$ if $\mathcal{G}_{\leftrightarrow}[X]$ is connected.*

**Definition 4** (Hedge [16]). *Let $\mathcal{G}$ be an ADMG, and $Y \subsetneq X$ be two subsets of its vertices, where $Y$ is*
108  *a district in $\mathcal{G}[Y]$. Vertices $X$ form a hedge for $Q[Y]$ if $X$ is a district in $\mathcal{G}[X]$ and $Anc_{\mathcal{G}[X]}(Y) = X$[1].*

**Definition 5** (Maximal hedge [1]). *For ADMG $\mathcal{G}$ and a set of its vertices $Y$, let $X$ be the union of all*
110  *hedges formed for $Q[Y]$. Graph $\mathcal{G}[X]$, denoted by $\boldsymbol{MH}(\mathcal{G}, Y)$, is called the maximal hedge for $Q[Y]$.*

111  As an example, both sets $\{t, x\}$ and $\{z, x\}$ form a hedge for $Q[x]$ in $\mathcal{G}$ in Figure 1a, and $\mathcal{G}[\{x, z, t\}]$
112  is the maximal hedge for $Q[x]$.

### 2.1  Problem setup

114  Let $\mathcal{G} = (V^{\mathcal{G}}, E_d^{\mathcal{G}}, E_b^{\mathcal{G}})$ be an ADMG, where $V^{\mathcal{G}}$ is the set of vertices each representing an observed
115  variable of the system, $E_d^{\mathcal{G}}$ is the set of directed edges, and $E_b^{\mathcal{G}}$ is the set of bidirected edges among
116  $V^{\mathcal{G}}$. We know *a priori* that the true ADMG describing the system is an edge-induced subgraph of
117  $\mathcal{G}$,[2] and we are given a probability map that indicates for each subgraph of $\mathcal{G}$ such as $\mathcal{G}_s$, with what
118  probability $\mathcal{G}_s$ is the true causal ADMG of the system. We denote this probability as $P(\mathcal{G}_s)$. For
119  instance, if edge probabilities $p_e$ are assumed to be mutually independent, $P(\mathcal{G}_s)$ takes the form:

$$P(\mathcal{G}_s) = \prod_{e \in \mathcal{G}_s} p_e \prod_{e \notin \mathcal{G}_s} (1 - p_e). \tag{1}$$

120  In what follows, we will refer to $P(\mathcal{G}_s)$ simply as the probability of the ADMG $\mathcal{G}_s$. The first problem
121  of our interest is formally defined as follows.

**Problem 1.** *We consider the problem of finding the most probable edge-induced subgraph of $\mathcal{G}$, in*
123  *which the causal effect $Q[Y]$ is identifiable. That is, the goal is to find the ADMG $\mathcal{G}^*$ defined by*

$$\mathcal{G}^* := \underset{\substack{\mathcal{G}_s \subseteq \mathcal{G}, \\ \mathcal{G}_s \in [\mathcal{G}]_{Id(Q[Y])}}}{\arg\max} \; P(\mathcal{G}_s). \tag{2}$$

---

[1]Akbari et al. [1] showed that this intuitive definition is equivalent to the standard definition of hedge in [16].
[2]Note that $\mathcal{G}$ can be a complete graph over both its directed and bidirected edges.

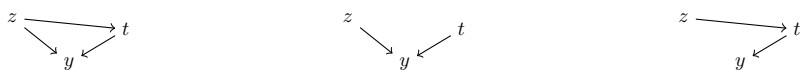

Figure 2: Three different graphs that share the same set $Anc_{\mathcal{G}}(\{y\}) = \{z, t\}$.

We will prove in Proposition 1 that if $Q[Y]$ is identifiable in $\mathcal{G}$, then it is also identifiable in every edge-induced subgraph of $\mathcal{G}$. In other words, if $\mathcal{G}$ is a feasible solution to the above optimization problem, so are all its edge-induced subgraphs. Furthermore, the same identification functional that is valid w.r.t. $\mathcal{G}$, is also valid w.r.t. every subgraph of $\mathcal{G}$. Let us illustrate this first on an example.

**Example 1.** *Consider the ADMG in Figure 1a. With the given edge probabilities and assuming independence among the edge probabilities, the subgraph of $\mathcal{G}$ illustrated in Figure 1b has probability $0.7 \times 0.7 \times 0.1 = 0.049$, whereas the subgraph of Figure 1c has probability $0.3 \times 0.3 \times 0.9 = 0.081$ (see Eq. (1)). If we were to solve Problem 1, we would choose $\mathcal{G}_2$ over $\mathcal{G}_1$, as it has a higher probability. Now consider identification formulas in $\mathcal{G}_1$ and $\mathcal{G}_2$, respectively:*

$$\mathcal{F}_1: \quad Q[Y] = P(Y|X), \quad \mathcal{F}_2: \quad Q[Y] = \sum_{Z,T} P(Y|X, Z, T) P(Z, T).$$

*$\mathcal{F}_1$ is a valid identification formula for any edge-induced subgraph of $\mathcal{G}_1$ (see Proposition 1). Analogously, $\mathcal{F}_2$ is valid for all edge-induced subgraphs of $\mathcal{G}_2$. If we consider the aggregate probability of the subgraphs of $\mathcal{G}_1$ and $\mathcal{G}_2$, i.e.,*

$$\sum_{\hat{\mathcal{G}} \subseteq \mathcal{G}_1} P(\hat{\mathcal{G}}) = 1 - 0.9 = 0.1, \quad \text{versus} \quad \sum_{\hat{\mathcal{G}} \subseteq \mathcal{G}_2} P(\hat{\mathcal{G}}) = (1 - 0.7) \times (1 - 0.7) = 0.09,$$

*then we should prefer choosing $\mathcal{G}_1$ over $\mathcal{G}_2$, as its identification formula $\mathcal{F}_1$ is more likely to be valid than $\mathcal{F}_2$ considering the fact that for all its subgraphs, the identification functional $\mathcal{F}_1$ is still valid.*

Plausibility of a certain identification functional $\mathcal{F}$ is the sum of the probabilities of all graphs in which $\mathcal{F}$ is valid given the query of interest. Finding the most plausible identification formula for a given query requires computing the plausibility of all formulae. Since the space of all formulae is intractable, an alternative approach to solve this problem is enumerating all valid formulae for a given graph. This changes the search space of the problem to the space of all graphs. However, this is yet another challenging and to the best of our knowledge open problem. Therefore, we propose the following problem as a surrogate that maximizes a lower bound of the most plausible identification formula. To do so, we use the result of Proposition 1 that shows when an identification functional is valid in a causal graph, it is also valid in all its edge-induced subgraphs.

**Problem 2.** *Consider the problem of finding the edge-induced subgraph $\mathcal{H}^*$ of $\mathcal{G}$ with maximum aggregate probability of its subgraphs, in which $Q[Y]$ is identifiable. Formally,*

$$\mathcal{H}^* := \underset{\mathcal{G}_s \subseteq \mathcal{G},\, \mathcal{G}_s \in [\mathcal{G}]_{Id(Q[Y])}}{\arg\max} \sum_{\hat{\mathcal{G}} \subseteq \mathcal{G}_s} P(\hat{\mathcal{G}}). \tag{3}$$

In other words, we are looking for a graph $\mathcal{H}^*$ with the maximum aggregate probability of its subgraphs, among the graphs in $[\mathcal{G}]_{Id(Q[Y])}$, i.e., the graphs in which $Q[Y]$ is identifiable. Running an identification algorithm (such as the ID function of [16]) on $\mathcal{H}^*$ yields an identification formula for $Q[Y]$ which is valid at least with the aggregate probability of the subgraphs of $\mathcal{H}^*$. Therefore, Problem 2 is a surrogate to recovering the identification formula with the highest plausibility.

In the sequel, for simplicity, we study Problems 1 and 2 under the following assumption. However, as proved in Appendix C, our results are valid in a more general setting where we allow only for perfect negative or positive correlations among the edges. An example of perfect negative correlation between two edges is that both of them cannot exist simultaneously. Appendix C.1 discusses the significance of this generalization.

**Assumption 1.** *The edges of $\mathcal{G}$ are mutually independent. That is, the probability of a subgraph $\mathcal{G}_s$ of $\mathcal{G}$ is of the form in (1).*

**Remark 1.** *It is noteworthy that our results are not limited to causal queries of the form $Q[Y] = P(Y|do(V^{\mathcal{G}} \setminus Y))$. They can be applied to general causal queries of the form $P_X(Y)$ if the set $Anc_{\mathcal{G} \setminus X}(Y)$ is known. This is because the causal query $P_X(Y)$ can be expressed as*

164   $\sum_{Anc_{\mathcal{G} \setminus X}(Y) \setminus Y} Q[Anc_{\mathcal{G} \setminus X}(Y)]$, *where $Anc_{\mathcal{G} \setminus X}(Y)$ is the set of ancestors of $Y$ in $\mathcal{G}$ after removing*
165   *the vertices of $X$. Furthermore, $P_X(Y)$ is identifiable in $\mathcal{G}$ if and only if $Q[Anc_{\mathcal{G} \setminus X}(Y)]$ is iden-*
166   *tifiable in $\mathcal{G}$ [19, 16, 9]. Note that the assumption that $Anc_{\mathcal{G} \setminus X}(Y)$ is known is not equivalent to*
167   *precluding uncertainty on the directed edges (as in the case of fixing the edge probabilities to 0 or 1),*
168   *but it rather imposes a perfect correlation type of constraint. Consider for instance the three graphs*
169   *of Figure 2, where all of them share the same set $Anc_{\mathcal{G} \setminus X}(Y) = \{z, t\}$. In fact, knowing this set*
170   *forces a constraint of the type that if the edge $z \to y$ does not exist, the path $z \to t \to y$ must.*

## 3   Reduction to Edge ID problem and establishing complexity

172   We begin this section with the following proposition, to which we referred before. Thereafter, we
173   discuss the hardness of the two problems considered in this work.

174   **Proposition 1.** *For any causal query $P_X(Y)$ and ADMG $\mathcal{G}$, if $\mathcal{F}$ is a valid identification formula for*
175   *$P_X(Y)$ in $\mathcal{G}$ (Def. 2), then $\mathcal{F}$ is a valid identification formula for $P_X(Y)$ in any $\mathcal{G}' \subseteq \mathcal{G}$.*

176   All proofs are presented in Appendix A. In what follows, we first formally define the edge ID problem,
177   and then show the equivalence of Problems 1 and 2 to the edge ID problem under Assumption 1.

178   **Definition 6** (Edge ID problem). *For ADMG $\mathcal{G} = (V^{\mathcal{G}}, E_d^{\mathcal{G}}, E_b^{\mathcal{G}})$, a set of non-negative edge weights*
179   *$W_{\mathcal{G}} = \{w_e \geq 0 | e \in \mathcal{G}\}$, and a causal query $Q[Y]$ for a subset of variables $Y \subseteq V^{\mathcal{G}}$, the objective*
180   *of the edge ID problem is to find the set of edges $E^* \subseteq E_d^{\mathcal{G}} \cup E_b^{\mathcal{G}}$ with minimum aggregated weight*
181   *(cost), such that $Q[Y]$ is identifiable in the graph resulting from removing $E^*$ from $\mathcal{G}$. Formally,*

$$E^* := \underset{E \subseteq E_d^{\mathcal{G}} \cup E_b^{\mathcal{G}}}{\arg\min} \sum_{e \in E} w_e,$$
$$\text{s.t.} \quad \mathcal{G}' = (V^{\mathcal{G}}, E_d^{\mathcal{G}} \setminus E, E_b^{\mathcal{G}} \setminus E) \in [\mathcal{G}]_{Id(Q[Y])}. \tag{4}$$

182   *We implicitly assume that the cost of removing a set of edges from $\mathcal{G}$ is the sum of the weights of each*
183   *individual edge.*

184   The following result unifies the two problems considered in this work by establishing their equivalence
185   to the edge ID problem. This is done by transforming Problems 1 and 2 with multiplicative objectives
186   into the edge ID problem that has an additive objective.

187   **Lemma 1.** *Under Assumption 1, Problem 1 is equivalent to the edge ID problem with the edge*
188   *weights chosen to be the log propensity ratios, i.e., $w_e = \max\{0, \log(\frac{p_e}{1-p_e})\}$, $\forall e \in \mathcal{G}$. Moreover,*
189   *Problem 2 is equivalent to the edge ID problem with the choice of weights $w_e = -\log(1 - p_e)$,*
190   *$\forall e \in \mathcal{G}$. That is, an instance of Problems 1 and 2 can be reduced to an instance of the edge ID*
191   *problem in polynomial time, and vice versa.*

192   As we mentioned earlier, the equivalence of these three problems can be established in more general
193   settings than what is described under Assumption 1. We refer the interested reader to Appendix C for
194   a discussion on one such setting. The following result shows that no polynomial-time algorithm for
195   solving any of these three problems exists unless P = NP.

196   **Theorem 1.** *The edge ID problem is NP-hard.*

197   Theorem 1 is established through a reduction from the minimum vertex cover problem, which is
198   known to be NP-hard [11]. Theorem 1 is a key result which shows the hardness of recovering the
199   most plausible graph in which a specified causal effect of interest is identifiable.

200   **Corollary 1.** *Problems 1 and 2 are NP-hard under Assumption 1.*

201   It is noteworthy that the size of the problem depends on the number of vertices of $\mathcal{G}$, i.e., $|V^{\mathcal{G}}|$, and
202   the number of edges of $\mathcal{G}$ with finite weight, i.e., $|E^{\mathcal{G}}| = |E_d^{\mathcal{G}}| + |E_b^{\mathcal{G}}|$. Since the ID algorithm
203   (function ID of [16]) runs in time $\mathcal{O}(|V^{\mathcal{G}}|^2)$, the brute-force algorithm that tests the identifiability of
204   $Q[Y]$ in every edge-induced subgraph of $\mathcal{G}$ and chooses the one with the minimum weight of deleted
205   edges runs in time $\mathcal{O}(2^{|E^{\mathcal{G}}|}|V^{\mathcal{G}}|^2)$. In the next Section, we present various algorithmic approaches
206   for solving or approximating the solutions to these problems.

---

**Algorithm 1** Recursive Algorithm for edge ID.

---

1: **function EDGEID**$(\mathcal{G}, Y, W_{\mathcal{G}}, \omega^{ub}, \omega^{th})$
2:     $\mathcal{H} \leftarrow \textbf{MH}(\mathcal{G}, Y)$
3:     **if** $\mathcal{H} = \mathcal{G}[Y]$ **then return** $(True, \emptyset)$
4:     $ID \leftarrow False, E_{min} \leftarrow \emptyset$
5:     **while** True **do**
6:         $e \leftarrow$ The edge of $\mathcal{H}$ with minimum weight
7:         **if** $w_e = \infty$ **or** $w_e > \omega^{ub}$ **then return** $(ID, E_{min})$
8:         $(id, E) \leftarrow \textbf{EDGEID}(\mathcal{H} \setminus e, Y, W_{\mathcal{G}} \setminus \{w_e\}, \omega^{ub} - w_e, \omega^{th} - w_e)$
9:         **if** $id = True$ **then**
10:           $ID \leftarrow True, \omega_E \leftarrow w_e + \sum_{e_j \in E} w_{e_j}$
11:           $\omega^{ub} \leftarrow \omega_E, E_{min} \leftarrow E \cup \{e\}$
12:           **if** $\omega^{ub} \leq \omega^{th}$ **then return** $(ID, E_{min})$
13:         Update $w_e \leftarrow \infty$ in $W_{\mathcal{G}}$

---

## 4 Algorithmic approaches

We first present a recursive approach for solving the edge ID problem in Section 4.1, described in Algorithm 1. Since the problem itself is NP-hard, Algorithm 1 runs in exponential time in the worst case. In Section 4.2, we present heuristic approximations of the edge ID problem which run in cubic time in the worst case. These heuristics can also be used as a pre-process to reduce the runtime of Alg. 1 by providing an upper bound which can be fed into Alg. 1 to prune the search space. Finally, in Section 4.3, we present a reduction of edge ID to yet another NP-hard problem, namely minimum-cost intervention problem [1], which allows us to use the algorithms designed for that problem to solve edge ID. Our simulations in Section 5 evaluate these approaches against each other.

### 4.1 Recursive exact algorithm

This approach is described in Algorithm 1. The inputs to the algorithm are an ADMG $\mathcal{G}$ along with edge weights $W_{\mathcal{G}}$, a set of vertices $Y$ corresponding to the causal query $Q[Y]$, an upper bound $\omega^{ub}$ on the aggregate weight (cost) of the optimal solution, and a threshold $\omega^{th}$, an upper bound on the acceptable cost of a solution. The closer $\omega^{ub}$ is to the optimal cost, the quicker Algorithm 1 will find the solution. If no upper bound is known, the algorithm can be initiated with $\omega^{ub} = \infty$. However, we shall discuss a few approaches to find a good upper bound $\omega^{ub}$ in the following Section. Note that when $\omega^{th} = 0$, Algorithm 1 will output the optimal solution. Otherwise, as soon as a feasible solution with weight less than $\omega^{th}$ is found, the algorithm terminates (line 12).

The algorithm begins with calling subroutine **MH** in line 2, which constructs the maximal hedge for $Q[Y]$, denoted by $\mathcal{H}$. We discuss this subroutine in detail in Appendix B. Throughout the rest of the algorithm, we only consider the edges in $\mathcal{H}$, as the other edges do not alter the identifiability. If there is no hedge formed for $Q[Y]$, i.e., $\mathcal{H} = \mathcal{G}[Y]$, there is no need to remove any edges from $\mathcal{G}$ and the effect is already identified. Otherwise, we remove the edge with the lowest weight ($e$) from $\mathcal{H}$ and recursively call the algorithm to find the solution after removing the edge $e$, unless the weight of $e$ is already higher than the upper bound $\omega^{ub}$, which means no feasible solutions exist for the provided upper bound (line 7). Whenever a feasible solution is found, the upper bound $\omega^{ub}$ is updated to the lowest weight among all the solutions weights discovered so far (line 11). This in turn helps the algorithm prune the exponential search space during the next iterations to reduce the runtime. As soon as a solution with a weight less than the acceptable threshold, i.e., $\omega^{th}$, is found, the algorithm returns the solution. Otherwise, $w_e$ is updated to infinity so that it never gets removed (line 13). This is due to the fact that we have already explored all the solutions involving $e$.

### 4.2 Heuristic algorithms

In this Section, we present two heuristic algorithms for approximating the solution to the edge ID problem. These algorithms can also be used to find the upper bound $\omega^{ub}$ efficiently, which could be fed as an input to Algorithm 1.

Let $Z = \{z \in V^{\mathcal{G}} | \exists y \in Y : \{z, y\} \in E_b^{\mathcal{G}}\} \setminus Y$ denote the set of vertices that have at least one common bidirected edge with a vertex in $Y$. Any hedge formed for $Q[Y]$ contains at least one vertex

---

**Algorithm 2** Heuristic algorithm for Edge ID.

---

1: **function HEID**$(\mathcal{G}, Y, W_{\mathcal{G}})$
2:     $\mathcal{G}' \leftarrow \mathbf{MH}(\mathcal{G}, Y)$ , $Z \leftarrow \{z \in V^{\mathcal{G}'} | \exists y \in Y : \{z, y\} \in E_b^{\mathcal{G}'}\} \setminus Y$
3:     $\mathcal{H} \leftarrow$ The induced subgraph of $\mathcal{G}'$ on its directed edges.
4:     $W_{\mathcal{H}} \leftarrow \{w_e \in W_{\mathcal{G}} | e \in \mathcal{H}\}$, $V^{\mathcal{H}} \leftarrow V^{\mathcal{H}} \cup \{y^*, z^*\}$
5:     **for** $z \in Z$ **do** $E^{\mathcal{H}} \leftarrow E^{\mathcal{H}} \cup (z^*, z)$, $W_{\mathcal{H}} \leftarrow W_{\mathcal{H}} \cup \{w_{(z^*, z)} = \sum_y w_{\{z, y\}}\}$
6:     **for** $y \in Y$ **do** $E^{\mathcal{H}} \leftarrow E^{\mathcal{H}} \cup (y, y^*)$, $W_{\mathcal{H}} \leftarrow W_{\mathcal{H}} \cup \{w_{(y, y^*)} = \infty\}$
7:     $E \leftarrow MinCut(\mathcal{H}, W_{\mathcal{H}}, z^*, y^*)$
8:     **return** $(E, \sum_{e \in E} w_e)$

---

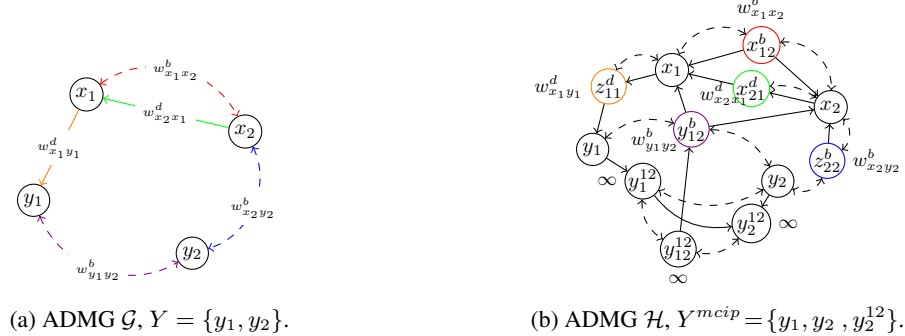

(a) ADMG $\mathcal{G}$, $Y = \{y_1, y_2\}$.          (b) ADMG $\mathcal{H}$, $Y^{mcip} = \{y_1, y_2, y_2^{12}\}$.

Figure 3: Reduction from edge ID to MCIP.

of $Z$. As a result, in order to eliminate all the hedges formed for $Q[Y]$, it suffices to make sure that none of the vertices in $Z$ appear in such a hedge. To this end, for any $z \in Z$, it suffices to either remove all the bidirected edges between $z$ and $Y$, or eliminate all the directed paths from $z$ to $Y$. The problem of eliminating all directed paths from $Z$ to $Y$ can be cast as a minimum cut problem between $Z$ and $Y$ in the edge-induced subgraph of $\mathcal{G}$ over its directed edges. To add the possibility of removing the bidirected edges between $Z$ and $Y$, we add an auxiliary vertex $z^*$ to the graph, and draw a directed edge from $z^*$ to every $z \in Z$ with weight $w = \sum_{y \in Y} w_{\{z, y\}}$, i.e., the sum of the weights of all bidirected edges between $z$ and $Y$. Note that $z$ can have bidirected edges to multiple vertices in $Y$. We then solve the minimum cut problem for $z^*$ and $Y$. If an edge between $z^*$ and $z \in Z$ is included in the solution to this minimum cut problem, it is mapped to removing all the bidirected edges between $z$ and $Y$ in the main problem. Note that we can run the algorithm on the maximal hedge formed for $Q[Y]$ in $\mathcal{G}$ rather than $\mathcal{G}$ itself. This heuristic is presented as Algorithm 2.

An analogous approach which goes through solving an undirected minimum cut on the edge induced subgraph of $\mathcal{G}$ over its bidirected edges is presented in Algorithm 4 in Appendix D. As mentioned earlier, these algorithms can be used either as standalone algorithms to approximate the solution to the edge ID problem, or as a pre-processing step to find an upper bound $\omega^{ub}$ for Algorithm 1. As we shall see in our simulations, both algorithms achieve near-optimal results on random graphs.

### 4.3 Alternative approach: reduction to MCIP

As an alternative approach to the algorithms discussed so far, we present a reduction of the edge ID problem to another NP-hard problem, i.e., the minimum-cost intervention problem (*MCIP*) introduced in [1]. This reduction allows us to exploit algorithms designed for MCIP to solve our problems. The formal definition of MCIP is as follows.

**Definition 7** (MCIP). *Suppose $\mathcal{G} = (V^{\mathcal{G}}, E_d^{\mathcal{G}}, E_b^{\mathcal{G}})$ is an ADMG, $C : V^{\mathcal{G}} \to \mathbb{R}^{\geq 0}$ is a cost function mapping each vertex of $\mathcal{G}$ to a non-negative cost, and $Y \subseteq V^{\mathcal{G}}$. The objective of the minimum-cost intervention problem for identifying the causal effect $Q[Y]$ is to find the subset $A \subseteq V^{\mathcal{G}}$ with the minimum aggregate cost such that $Q[Y]$ is identifiable after intervening on the set $A$.*

The reduction from edge ID to MCIP is based on a transformation from ADMG $\mathcal{G}$ to another ADMG $\mathcal{H}$, where each edge in $\mathcal{G}$ is represented by a vertex in $\mathcal{H}$. This transformation is based on the causal

query $Q[Y]$, and it maps the identifiability of $Q[Y]$ in $\mathcal{G}$ to identifiability of $Q[Y^{mcip}]$ in $\mathcal{H}$, where $Y^{mcip}$ is a set of vertices in $\mathcal{H}$. This transformation satisfies the following property; removing a set of edges $E^*$ in $\mathcal{G}$ makes $Q[Y]$ identifiable if and only if intervening on the corresponding vertices of $E^*$ in $\mathcal{H}$ makes $Q[Y^{mcip}]$ identifiable. More precisely, after this transformation, solving the edge ID problem for $Q[Y]$ in $\mathcal{G}$ is equivalent to solving MCIP for $Q[Y^{mcip}]$ in $\mathcal{H}$. The complete details of this transformation can be found in Appendix A.2. An example of this reduction is shown in Figure 3, where $Q[\{y_1, y_2\}]$ in $\mathcal{G}$ (Figure 3a) is mapped to $Q[\{y_1, y_2, y_2^{12}\}]$ in $\mathcal{H}$ (Figure 3b), where $\{y_1, y_2, y_2^{12}\}$ is a district, and the set of all vertices of $\mathcal{H}$ forms a hedge for it. The vertices of $\mathcal{H}$ corresponding to each edge in $\mathcal{G}$ are indicated with the same color and have the same weight (cost). To avoid intervening on the remaining vertices in $\mathcal{H}$, we assign infinity cost to them. It is straightforward to see that the solution to the edge ID problem in $\mathcal{G}$ with the query $Q[Y = \{y_1, y_2\}]$ would be to remove the edge with the lowest weight. This is because after removing any edge in $\mathcal{G}$, no hedge remains for $Q[Y]$. Similarly, in $\mathcal{H}$, the solution to MCIP with the query $Q[Y^{mcip} = \{y_1, y_2, y_2^{12}\}]$ is to intervene on the vertex with the lowest cost among $Z = \{z_{11}^d, x_{21}^d, x_{12}^b, y_{12}^b, z_{22}^b\}$. This is because after intervening on any vertex in $Z$, no hedge remains for $Q[Y^{mcip}]$. The following result formally establishes the link between the edge ID problem in $\mathcal{G}$ and MCIP in $\mathcal{H}$.

**Proposition 2.** *There exists a polynomial-time reduction from edge ID to MCIP and vice versa.*

## 5 Experiments

We evaluate the proposed heuristic algorithms 2 (HEID-1) and 4 (HEID-2), as well as the exact algorithm 1 (EDGEID), where the upper-bound $\omega^{ub}$ for EDGEID is set to be the minimum cost found by HEID-1 or -2. Furthermore, given the reduction of the edge ID problem to the MCIP problem described in Section 4.3, we also evaluate the two approximation and one exact algorithms from [1] (MCIP-H1, MCIP-H2, and MCIP-exact, respectively). Experimental results are provided for Problem 1, and analogous results for Problem 2 are provided in Appendix E.3. All experiments were carried out on an Intel i9-9900K CPU running at 3.6GHz.

**Simulations:** The algorithms are evaluated for graphs with between 5 and 250 vertices. For a given number of vertices, we uniformly sample 50 ADMG structures from a library of graphs which are non-isomorphic to each other. Edges for each of these 100 graphs are sampled with probability of $\log(n)/n$, where $n$ is the number of (observable) vertices, to impose sparsity (thus pragmatically reducing the search space). For each graph we sample directed and bidirected edge probabilities $p_e$ uniformly between 0.51 and $1.0$[3]. The problem is then converted into edge ID according to Lemma 1. The vertices in the graphs are topologically sorted and the outcome $Y$ is selected to be the last vertex in the topological ordering. We then check whether a solution exists in principle by removing all finite cost edges and checking for identifiability. If not, a new graph is sampled to avoid evaluating the algorithms on graphs with no solution. For each of these 50 probabilistic ADMGs, we run the algorithms and record the resulting runtime and the associated cost of the solution. If the runtime exceeds 3 minutes, we abort and log that the algorithm has failed to find a solution.

Results are presented in Figure 4. Runtimes and costs are shown for the subset of graphs for which all algorithms found a solution (to facilitate comparison). Runtimes for each algorithm are shown in Fig. 4a, where it can be seen that our proposed HEID-1 and HEID-2 heuristic algorithms have negligible runtime, followed by the MCIP variants. Interestingly, the exact algorithm EDGEID outperformed the MCIP algorithms on larger graphs, for which the transformation time from the edge ID problem to the MCIP increases with the size of the graph. In contrast, EDGEID had large runtime variance which depended heavily on the specifics of the graph under evaluation, particularly for graphs with fewer vertices. The costs for each graph are shown in Fig. 4b, and here we see, as expected, the lowest cost is achieved by the two exact algorithms, EDGEID and MCIP-exact, followed closely by the heuristic algorithms. Figure 4c shows the fraction of evaluations for which the runtime exceeded 3 minutes (applicable to the exact algorithms). In general, and owing to the sparsity penalty in our graph generation mechanism, the cost of identified solutions falls with the number of vertices. However, among the exact algorithms, EDGEID, exceeds the 3 minute runtime more often than the MCIP-Exact, regardless of the number of vertices and despite the fact that EDGEID is quicker at finding a solution when it does so. Overall, HEID-1 was both the most consistent in terms of finding a solution, having a short runtime, and achieving a close to optimal cost.

---

[3]Note that we do not consider edge probabilities less than 0.5 as from Lemma 1, such edges would be mapped to edges with 0 weight in the equivalent edge ID problem, which can always be removed at the beginning.

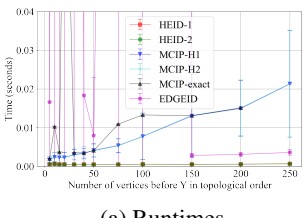 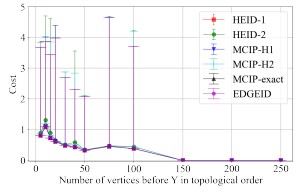 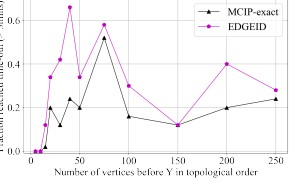

| (a) Runtimes. | (b) Solution costs. | (c) Fraction runtime exceeded 3 min. |

Figure 4: Experimental results for runtime, solution costs, fraction of graphs for which no solution was found, and fraction of graphs for which runtime limit of 3 minutes was exceeded. Error bars for runtime and cost graphs indicate 5th and 95th percentiles. Best viewed in color.

**Real-World Graphs:** We also apply the algorithms to four real-world datasets. The first 'Psych' (22 nodes & 70 directed edges) concerns the putative structure from a causal discovery algorithm Structural Agnostic Model [10] using data collected as part of the Health and Relationships Project [20]. The other three 'Barley' (48 nodes & 84 directed edges), 'Water' (32 nodes & 66 directed edges), and 'Alarm' (37 nodes & 46 directed edges) come from the bnlearn python package [17]. For all four graphs, and as with the simulations described above, we introduce bidirected edges with a sparsity constraint of $\log(n)/n$, and simulate expert domain knowledge by random assigning directed and bidirected edge probabilities between 0.51 and 1. The outcome $Y$ is selected to be the last vertex in the topological ordering. For these results, we provide the runtime (limited to 500 seconds) and cost, as well as the ratio of graph plausibility before and after selecting a subgraph in which the effect is identifiable $P(\hat{\mathcal{G}}^*)/P(\mathcal{G})$. This ratio is 1.0 if the effect is identifiable in the original graph, and decreases according to the plausibility of an identified subgraph.

Results are shown in Table 1. In cases where MCIP-exact and/or EDGEID did not exceed the runtime limit, it can be seen that HEID-2 and MCIP-H2 achieved equivalent to optimal cost and ratio. Runtimes for MCIP variants exceeded the HEID variants owing to the required transformation. EDGEID timed out on all but the Alarm structure, whereas MCIP-exact only timed out on the Psych structure, indicating that the MCIP-exact is more consistent (this also corroborates Figure 4c).

Table 1: Time (seconds), cost, and ratio $P(\hat{\mathcal{G}}^*)/P(\mathcal{G})$ for seven algorithms over four real-world datasets. A dash - indicates maximum runtime (500 seconds) exceeded.

| Algorithm | Psych | | | Barley | | | Alarm | | | Water | | |
|---|---|---|---|---|---|---|---|---|---|---|---|---|
| | Time | Cost | Ratio | Time | Cost | Ratio | Time | Cost | Ratio | Time | Cost | Ratio |
| HEID-1 | 0.0019 | 2.648 | 0.07 | 0.0026 | 0.081 | 0.92 | 0.0004 | 0.0 | 1.0 | 0.0019 | 1.02 | 0.36 |
| HEID-2 | 0.0019 | 1.806 | 0.16 | 0.0026 | 0.081 | 0.92 | 0.0003 | 0.0 | 1.0 | 0.0017 | 0.42 | 0.66 |
| MCIP-H1 | 0.0136 | 2.648 | 0.07 | 0.0140 | 0.081 | 0.92 | 0.0027 | 0.0 | 1.0 | 0.0124 | 1.02 | 0.36 |
| MCIP-H2 | 0.0133 | 1.806 | 0.16 | 0.0131 | 0.081 | 0.92 | 0.0029 | 0.0 | 1.0 | 0.0113 | 0.42 | 0.66 |
| MCIP-exact | - | - | - | 0.0099 | 0.081 | 0.92 | 0.0028 | 0.0 | 1.0 | 0.0221 | 0.42 | 0.66 |
| EDGEID | - | - | - | - | - | - | 0.0005 | 0.0 | 1.0 | - | - | - |

# 6 Conclusion

Researchers in causal inference are often faced with graphs for which the effect of interest is not identifiable. It is common to identify a target effect by assuming ignorability. A less drastic and more reasonable approach would be to relax this assumption by identifying the most plausible subgraph, given uncertainty about the structure as we suggested in this work. We presented a number of algorithms for finding the most probable/plausible probabilistic ADMG in which the target causal effect is identifiable. We provided an analysis of the complexity of the problem, and an experimental comparison of runtimes, solution costs, and failure rates. We noted that our heuristic algorithms, Alg. 2 and Alg. 4 performed remarkably well across all metrics. In terms of limitations, we made the assumption that the edges in $\mathcal{G}$ are mutually independent (Assumption 1). Future work should explore the case where this assumption does not hold. Finally, it is worth noting that the external validity of the derived subgraph (i.e., whether or not the subgraph is correctly specified with respect to the corresponding real-world process) is not guaranteed. As such, practitioners that use such approaches are encouraged to do so with caution, in particular for research involving human participants.

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

# Appendix

The appendices are organized as follows. Formal proofs of the results stated in the main text are presented in Section A. In Section B, we describe the algorithm to recover the maximal hedge formed for a certain query (Def. 5), which is used as a subroutine of Algorithm 1. A generalization of Assumption 1 is discussed in Section C. Section D provides further details of the heuristic algorithms discussed in the main text. Further evaluations and experimental conditions for our proposed algorithms are presented in Section E.

Table 2: Table of notations

| Symbol | Description |
|---|---|
| $V^{\mathcal{G}}$ | Vertices of $\mathcal{G}$ |
| $E_b^{\mathcal{G}}$ | The set of bidirected edges of $\mathcal{G}$ |
| $E_d^{\mathcal{G}}$ | The set of directed edges of $\mathcal{G}$ |
| $Anc_{\mathcal{G}}(X)$ | Ancestors of $X$ in $\mathcal{G}$ |
| $\mathcal{M}(\mathcal{G})$ | The set of the all compatible models with $\mathcal{G}$ |
| $p_e$ | Probability of edge $e$ |
| $w_e$ | Weight of edge $e$ |
| $P_X(Y)$ | Causal effect of $X$ on $Y$ |

## A  Formal Proofs

We begin with presenting the proofs of Proposition 1 and Lemma 1. Proofs of Theorem 1 and Proposition 2 appear at the end of Sections A.1 and A.2, respectively.

**Proposition 1.** *For any causal query $P_X(Y)$ and ADMG $\mathcal{G}$, if $\mathcal{F}$ is a valid identification formula for $P_X(Y)$ in $\mathcal{G}$ (Def. 2), then $\mathcal{F}$ is a valid identification formula for $P_X(Y)$ in any $\mathcal{G}' \subseteq \mathcal{G}$.*

*Proof.* Let $\mathcal{H} \subseteq \mathcal{G}$ be an arbitrary edge-induced subgraph of $\mathcal{G}$. Let $\mathcal{F}$ be an identification formula for $P_X(Y)$ in $\mathcal{G}$, i.e., for any model $M$ that induces $\mathcal{G}$,

$$P_X^M(Y) = \mathcal{F}(P^M(V^{\mathcal{G}})). \tag{5}$$

By definition, $P_X(Y)$ is identifiable in $\mathcal{G}$. As a result, there exists and identification formula such as $\mathcal{F}'$ that can be derived for $P_X(Y)$ in $\mathcal{G}$, using a sequence of do calculus rules and basic probability manipulations. Note that this means for any model $M$ that induces $\mathcal{G}$,

$$P_X^M(Y) = \mathcal{F}'(P^M(V^{\mathcal{G}})). \tag{6}$$

Note that an immediate corollary of Equations 5 and 6 is that for any model $M$ that induces $\mathcal{G}$,

$$\mathcal{F}(P^M(V^{\mathcal{G}})) = \mathcal{F}'(P^M(V^{\mathcal{G}})). \tag{7}$$

Now, we first show that this sequence of actions (combination of do calculus rules and probability manipulations) is valid in $\mathcal{H}$. Note that the basic probability manipulations are graph-independent. It only suffices to show that any applied do calculus rule w.r.t. $\mathcal{G}$ can also be applied w.r.t. $\mathcal{H}$. The validity conditions of all three do calculus rules are based on certain d-separations. As a result, it suffices to show that if a d-separation relation is valid in $\mathcal{G}$, it is also valid in $\mathcal{H}$. To do so, it suffices to show that if all paths between $Z_1$ and $Z_2$ are blocked in $\mathcal{G}$ given $W$, they are blocked in $\mathcal{H}$ too, for arbitrary disjoint sets of vertices $Z_1, Z_2, W \subseteq V^{\mathcal{G}}$. Take an arbitrary path, $p$, between $Z_1$ and $Z_2$ in $\mathcal{H}$. Since $\mathcal{H} \subseteq \mathcal{G}$, this path exists in $\mathcal{G}$. Since $Z_1$ and $Z_2$ are d-separated given $W$ in $\mathcal{G}$, the path $p$ is blocked by $W$. As a result, any path between $Z_1$ and $Z_2$ in $\mathcal{H}$ is blocked by $W$. Therefore, any do-calculus rule applied in $\mathcal{G}$, can also be applied in $\mathcal{H}$. Hence, $\mathcal{F}'$ is a valid identification formula for $P_X(Y)$. That is, for any model $M$ that induces $\mathcal{H}$,

$$P_X^M(Y) = \mathcal{F}'(P^M(V^{\mathcal{H}})). \tag{8}$$

Now note that any model $M$ that induces $\mathcal{H}$, i.e., is compatible with $\mathcal{H}$, is also compatible with $\mathcal{G}$. Also, $V^{\mathcal{G}} = V^{\mathcal{H}}$. As a result, from Equations 7 and 8, we know that for any model $M$ that induces $\mathcal{H}$,

$$P_X^M(Y) = \mathcal{F}(P^M(V^{\mathcal{H}})).$$

By definition, $\mathcal{F}$ is a valid identification formula for $P_X(Y)$ in $\mathcal{H}$. $\qquad\square$

**Lemma 1.** *Under Assumption 1, Problem 1 is equivalent to the edge ID problem with the edge weights chosen to be the log propensity ratios, i.e., $w_e = \max\{0, \log(\frac{p_e}{1-p_e})\}$, $\forall e \in \mathcal{G}$. Moreover, Problem 2 is equivalent to the edge ID problem with the choice of weights $w_e = -\log(1 - p_e)$, $\forall e \in \mathcal{G}$. That is, an instance of Problems 1 and 2 can be reduced to an instance of the edge ID problem in polynomial time, and vice versa.*

*Proof. Problem 1.* First consider an arbitrary graph $\mathcal{G}_1 \in [\mathcal{G}]_{Id(Q[Y])}$ such that $\mathcal{G}_1$ has an edge $e$ with $p_e < 1/2$. Let $G_2$ denote the graph $G_1$ after removing $e$. Proposition 1 implies that $\mathcal{G}_2 \in [\mathcal{G}]_{Id(Q[Y])}$. According to Equation 1, we have $P(\mathcal{G}_2) = \frac{1-p_e}{p_e} P(\mathcal{G}_1) > P(\mathcal{G}_1)$ (since $p_e < 1/2$). As a result, the solution $\mathcal{G}^*$ to Problem 1 (Eq. 2) has no edges with probability less than $1/2$. We can therefore rewrite Problem 1 as:

$$\mathcal{G}^* := \underset{\substack{\mathcal{G}_s \subseteq \mathcal{G}, \\ \mathcal{G}_s \in [\mathcal{G}]_{Id(Q[Y])}}}{\arg\max} P(\mathcal{G}_s) = \underset{\substack{\mathcal{G}_s \subseteq \mathcal{G}, \\ \mathcal{G}_s \in [\mathcal{G}]_{Id(Q[Y])}}}{\arg\max} P(\mathcal{G}_s) \quad \text{s.t.} \quad \forall e \in \mathcal{G}_s : p_e \geq \frac{1}{2}.$$

Or equivalently, we can always assume that we start with a graph $\mathcal{G}$ that has no edges with probability less than $1/2$, as otherwise we can remove all of those edges and the problem does not change. This indeed is equivalent to choosing weight (cost) $0$ for those edges in the equivalent edge ID problem. Now assuming that the edges have probability at least $1/2$,

$$\mathcal{G}^* = \underset{\substack{\mathcal{G}_s \subseteq \mathcal{G}, \\ \mathcal{G}_s \in [\mathcal{G}]_{Id(Q[Y])}}}{\arg\max} P(\mathcal{G}_s)$$

$$= \underset{\substack{\mathcal{G}_s \subseteq \mathcal{G}, \\ \mathcal{G}_s \in [\mathcal{G}]_{Id(Q[Y])}}}{\arg\max} \log(P(\mathcal{G}_s))$$

$$= \underset{\substack{\mathcal{G}_s \subseteq \mathcal{G}, \\ \mathcal{G}_s \in [\mathcal{G}]_{Id(Q[Y])}}}{\arg\max} \log(\prod_{e \in \mathcal{G}_s} p_e \prod_{e \notin \mathcal{G}_s} (1 - p_e))$$

$$= \underset{\substack{\mathcal{G}_s \subseteq \mathcal{G}, \\ \mathcal{G}_s \in [\mathcal{G}]_{Id(Q[Y])}}}{\arg\max} \sum_{e \in \mathcal{G}_s} \log(p_e) + \sum_{e \notin \mathcal{G}_s} \log(1 - p_e))$$

$$= \underset{\substack{\mathcal{G}_s \subseteq \mathcal{G}, \\ \mathcal{G}_s \in [\mathcal{G}]_{Id(Q[Y])}}}{\arg\max} \sum_{e \in \mathcal{G}_s} \log(p_e) + \sum_{e \notin \mathcal{G}_s} \log(1 - p_e)) + \sum_{e \in \mathcal{G}_s} \log(1 - p_e)) - \sum_{e \in \mathcal{G}_s} \log(1 - p_e))$$

Since $\sum_{e \notin \mathcal{G}_s} \log(1 - p_e)) + \sum_{e \in \mathcal{G}_s} \log(1 - p_e))$ is a constant value that does not depend on $\mathcal{G}_s$, it can be ignored in the maximization and we have:

$$\mathcal{G}^* = \underset{\substack{\mathcal{G}_s \subseteq \mathcal{G}, \\ \mathcal{G}_s \in [\mathcal{G}]_{Id(Q[Y])}}}{\arg\max} \sum_{e \in \mathcal{G}_s} \log(p_e) - \sum_{e \in \mathcal{G}_s} \log(1 - p_e))$$

$$= \underset{\substack{\mathcal{G}_s \subseteq \mathcal{G}, \\ \mathcal{G}_s \in [\mathcal{G}]_{Id(Q[Y])}}}{\arg\max} \sum_{e \in \mathcal{G}_s} \log(\frac{p_e}{1 - p_e})$$

$$= \underset{\substack{\mathcal{G}_s \subseteq \mathcal{G}, \\ \mathcal{G}_s \in [\mathcal{G}]_{Id(Q[Y])}}}{\arg\min} \sum_{e \notin \mathcal{G}_s} \log(\frac{p_e}{1 - p_e}).$$

From the formulation above, it is clear that if we assign the weight $w_e = \log(\frac{p_e}{1-p_e})$ to each edge $e \in E^{\mathcal{G}}$, we will have an instance of the edge ID problem. Note that for edges with probability higher than $1/2$, $\log(\frac{p_e}{1-p_e}) \geq 0$, and this assignment of edge weights satisfies the positivity requirement. For the opposite direction, note that the procedure explained above is reversible by the choice of probabilities $p_e = \frac{\exp(w_e)}{1+\exp(w_e)}$, which is a value between $1/2$ and $1$.

*Problem 2.* First note that under Assumption 1, for any graph $\mathcal{G}_s$,

$$\sum_{\hat{\mathcal{G}} \subseteq \mathcal{G}_s} P(\hat{\mathcal{G}}) = \prod_{e \notin \mathcal{G}_s} (1 - p_e)[\sum_{\hat{E} \subseteq E^{\mathcal{G}_s}} \prod_{e \in \hat{E}} p_e \prod_{e \notin \hat{E}} (1 - p_e)] = \prod_{e \notin \mathcal{G}_s} (1 - p_e).$$

This is because the inner summation goes over all the possible subsets of $E^{\mathcal{G}_s}$, and the summation adds up to 1. Therefore, we can rewrite Problem 2 (Eq. 3)as

$$
\begin{aligned}
\mathcal{H}^* &= \underset{\substack{\mathcal{G}_s \subseteq \mathcal{G}, \\ \mathcal{G}_s \in [\mathcal{G}]_{Id(Q[Y])}}}{\arg\max} \sum_{\hat{\mathcal{G}} \subseteq \mathcal{G}_s} P(\hat{\mathcal{G}}) \\
&= \underset{\substack{\mathcal{G}_s \subseteq \mathcal{G}, \\ \mathcal{G}_s \in [\mathcal{G}]_{Id(Q[Y])}}}{\arg\max} \prod_{e \notin \mathcal{G}_s} (1 - p_e) \\
&= \underset{\substack{\mathcal{G}_s \subseteq \mathcal{G}, \\ \mathcal{G}_s \in [\mathcal{G}]_{Id(Q[Y])}}}{\arg\max} \log\left( \prod_{e \notin \mathcal{G}_s} (1 - p_e) \right) \\
&= \underset{\substack{\mathcal{G}_s \subseteq \mathcal{G}, \\ \mathcal{G}_s \in [\mathcal{G}]_{Id(Q[Y])}}}{\arg\max} \sum_{e \notin \mathcal{G}_s} \log(1 - p_e) \\
&= \underset{\substack{\mathcal{G}_s \subseteq \mathcal{G}, \\ \mathcal{G}_s \in [\mathcal{G}]_{Id(Q[Y])}}}{\arg\min} \sum_{e \notin \mathcal{G}_s} -\log(1 - p_e).
\end{aligned}
$$

With the same reasoning as before, assigning the weights $w_e = -\log(1 - p_e)$ to each edge $e \in E^{\mathcal{G}}$, we end up with an instance of the edge ID problem. Note that again $0 \leq -\log(1 - p_e) \leq \infty$. It is noteworthy that this procedure is also reversible with the choice of edge probabilities $p_e = 1 - \exp(-w_e)$, which reduces the edge ID problem to an instance of Problem 2. Again note that $0 \leq 1 - \exp(-w_e) \leq 1$ for any non-negative $w_e$. □

## A.1   Reduction from MCIP to edge ID

**Theorem 1.** *The edge ID problem is NP-hard.*

To prove Theorem 1, we first present a polynomial-time reduction from MCIP to the edge ID problem. It has been shown that the minimum vertex cover problem can be reduced to MCIP in polynomial time [1]. Combining the two reductions, we show that there exists a polynomial-time redcution from the minimum vertex cover problem to the edge ID problem. Since the minimum vertex cover problem is known to be NP-hard [11], it follows that the edge ID problem is also NP-hard.

We propose the following reduction from MCIP to the edge ID problem. Assume we want to solve MCIP given ADMG $\mathcal{G} = (V^{\mathcal{G}}, E_d^{\mathcal{G}}, E_b^{\mathcal{G}})$, query $Q[Y]$, and the intervention costs $C(v)$ for $v \in V^{\mathcal{G}}$. We construct a graph, denoted by $\mathcal{H} = \mathcal{T}_1(\mathcal{G}, Y)$, through the following steps.

a. For every vertex $x \in V^{\mathcal{G}} \setminus Y$, add two vertices $x^1, x^2$ to $V^{\mathcal{H}}$.

b. For any bidirected edge $\{x, z\} \in E_b^{\mathcal{G}}$ where $x \in V^{\mathcal{G}} \setminus Y$ and $z \in V^{\mathcal{G}}$, add the bidirected edge $\{x^2, z^2\}$ to $E_b^{\mathcal{H}}$.

c. For any directed edge $(x, z) \in E_d^{\mathcal{G}}$ where $x \in V^{\mathcal{G}} \setminus Y$ and $z \in V^{\mathcal{G}}$, add the directed edge $(x^1, z^1)$ to $E_d^{\mathcal{H}}$.

d. For any bidirected edge $\{y_1, y_2\} \in E_b^{\mathcal{G}}$ where $y_1, y_2 \in Y$, add the bidirected edge $\{y_1, y_2\}$ to $E_b^{\mathcal{H}}$.

e. For every $x^1, x^2 \in V^{\mathcal{G}} \setminus Y$, draw the two edges $\{x^1, x^2\} \in E_b^{\mathcal{H}}$ and $(x^2, x^1) \in E_d^{\mathcal{H}}$. Furthermore, the weight of $\{x^1, x^2\}$ is $C(x)$.

f. The costs of the all other edges in $\mathcal{H}$ are assigned to be infinite.

With abuse of notation, for any vertex $x \in V^{\mathcal{G}} \setminus Y$, we define $\mathcal{T}_1(x) = \{x^2, x^1\} \in E_b^{\mathcal{H}}$, where $\{x^2, x^1\}$ is the bidirected edge in $\mathcal{H}$ that corresponds to $x$ in $\mathcal{G}$, and inherits the same weight (cost).

**Example 2.** *Consider graph $\mathcal{G}$ in Figure 5a. Vertices $x$ and $z$ are mapped to $x^1, x^2$, and $z^1, z^2$, respectively. Both a directed and a bidirected edge are drawn between these pairs. The bidirected edge $\{x^1, x^2\}$ is assigned the weight $C(x) = c_x$, and the bidirected edge $\{z^1, z^2\}$ is assigned the weight $C(z) = c_z$. Infinite weights are assigned to the rest of the edges in $\mathcal{H}$ (Figure 5b).*

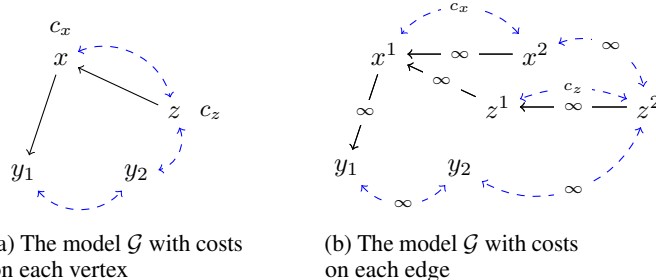

(a) The model $\mathcal{G}$ with costs
on each vertex

(b) The model $\mathcal{G}$ with costs
on each edge

Figure 5: Reduction of MCIP to edge ID

**Proposition 3.** *Suppose $\mathcal{G}'$ is an ADMG, $Y \subseteq V^{\mathcal{G}'}$ is a set of its vertices such that $Y$ is a district in $\mathcal{G}'[Y]$, and $\mathcal{H}' = \mathcal{T}_1(\mathcal{G}', Y)$. Consider $X \subseteq V^{\mathcal{G}'} \setminus Y$ as an arbitrary subset of vertices of $\mathcal{G}'$, and define $\mathcal{G} = \mathcal{G}'[V^{\mathcal{G}'} \setminus X]$. Let $E_b'' = \{e \in E_b^{\mathcal{H}'} | \exists v \in X, e = \mathcal{T}_1(v)\}$ and define $E_b^{\mathcal{H}} = E_b^{\mathcal{H}'} \setminus E_b''$. Let $\mathcal{H}$ be the edge-induced subgraph of $\mathcal{H}'$ defined as $\mathcal{H} = (V^{\mathcal{H}'}, E_d^{\mathcal{H}}, E_b^{\mathcal{H}})$. $Q[Y]$ is identifiable in $\mathcal{G}$ if and only if $Q[Y]$ is identifiable in $\mathcal{H}$.*

*Proof.* We prove the contrapositive, i.e., $Q[Y]$ is not identifiable in $\mathcal{G}$ iff $Q[Y]$ is not identifiable in $\mathcal{H}$. Note that by construction, $Y$ is a district in both $\mathcal{G}[Y]$ and $\mathcal{H}[Y]$. That is, it suffices to show that there exists a hedge formed for $Q[Y]$ in $\mathcal{G}$ iff there exists a hedge formed for $Q[Y]$ in $\mathcal{H}$.

To this end, we first prove the following claim. Let $W \in V^{\mathcal{H}}$ form a hedge for $Q[Y]$. If $x^1 \in W$ for some $x \in V^{\mathcal{G}'}$, then $x^2 \in W$ and vice versa. That is, the two vertices $x^1$ and $x^2$ corresponding to the same vertex $x$ in $V^{\mathcal{G}'}$ appear only simultaneously in any hedge. To see this, note that by construction, $x^1$ is the only child of $x^2$. By definition of hedge, if $x^2 \in W$, then it has a directed path to $Y$ within $\mathcal{H}[W]$, and this path can only go through $x^1$. For the other direction, note that $x^1$ has only one bidirected edge, which is with $x^2$. Again, by definition of hedge, if $x^1 \in W$, then it has a bidirected path to $Y$ within $\mathcal{H}[W]$, and this path can only go through $x^2$. Hence, in the sequel, when there is a hedge $W$ formed for $Q[Y]$ in $\mathcal{H}$, we will without loss of generality assume that there exists a set of variables $Z \subseteq V^{G'}$ such that $W = Z^1 \cup Z^2 \cup Y$, where $Z^1 = \{z^1 | z \in Z\}$ and $Z^2 = \{z^2 | z \in Z\}$.

*If part.* Let $W = Z^1 \cup Z^2 \cup Y$ form a hedge for $Q[Y]$ in $\mathcal{H}$. First note that since none of the bidirected edges between $Z^1$ and $Z^2$ are removed in $\mathcal{H}$, by construction, all vertices $Z$ are present in $\mathcal{G}$, i.e., $Z \subseteq V^{\mathcal{G}}$. Now we show that $Z \cup Y$ forms a hedge for $Q[Y]$ in $\mathcal{G}$. To this end, we prove $\mathcal{G}[Z \cup Y]$ is a district and $Z \cup Y = Anc_{\mathcal{G}[Z \cup Y]}(Y)$. First note that any vertex in $Z^1$ has only one bidirected edge to a vertex in $Z^2$. That is, if we consider the edge-induced subgraph of $\mathcal{H}[W]$ over its bidirected edges, vertices of $Z^1$ are leaf nodes. As a result, $Z^2 \cup Y$ must be connected in this graph. That is, $Z^2 \cup Y$ is a district in $\mathcal{H}[Z^2 \cup Y]$. This implies by construction of $\mathcal{H}$ that $\mathcal{G}[Z \cup Y]$ is a single district. With a similar reasoning, note that vertices in $Z^2$ have no parents. As result, $Z^1 \cup Y = Anc_{\mathcal{H}[Z^1 \cup Y]}(Y)$ (since the directed paths cannot go through $Z^2$). Again, by construction, the edge-induced subgraph of $\mathcal{G}[Z \cup Y]$ over its directed edges is a copy of $\mathcal{H}[Z^1 \cup Y]$. As a result, $Z \cup Y = Anc_{\mathcal{G}[Z \cup Y]}(Y)$.

*Only if part.* Let $Z \cup Y$ form a hedge for $Q[Y]$ in $\mathcal{G}$, where $Z \subseteq V^{\mathcal{G}} \setminus Y$. Define $Z^1 = \{z^1 | z \in Z\}$ and $Z^2 = \{z^2 | z \in Z\}$. We show that $Z^1 \cup Z^2 \cup Y$ forms a hedge for $Q[Y]$ in $\mathcal{H}$. First, by definition of hedge, $Anc_{\mathcal{G}[Z \cup Y]}(Y) = Z \cup Y$. Since the edge-induced subgraph of $\mathcal{H}[Z^1 \cup Y]$ is a copy of $\mathcal{G}[Z \cup Y]$ by construction, we know $Anc_{\mathcal{G}[Z^1 \cup Y]}(Y) = Z^1 \cup Y$. Further, each vertex $z^2 \in Z^2$ is a parent of $z^1 \in Z^1$. As a result, $Anc_{\mathcal{G}[Z^1 \cup Z^2 \cup Y]}(Y) = Z^1 \cup Z^2 \cup Y$. Now it suffices to show that $Z^1 \cup Z^2 \cup Y$ is a district in $\mathcal{H}[Z^1 \cup Z^2 \cup Y]$. By definition of hedge, $Z \cup Y$ is a district in $\mathcal{G}[Z \cup Y]$. By construction of $\mathcal{H}$, exactly the same bidirected edges (and therefore bidirected paths) exist in $\mathcal{H}[Z^2 \cup Y]$. Therefore, $Z^2 \cup Y$ is a district in $\mathcal{H}[Z^2 \cup Y]$. Now note that by construction of $\mathcal{H}'$, each vertex $z^1 \in Z^1$ has a bidirected edge to $z^2 \in Z^2$. And by definition of $\mathcal{G}$ and $\mathcal{H}$, since the vertices $Z$ exist in $\mathcal{G}$, none of these edges are removed in $\mathcal{H}$. As a result, $Z^1 \cup Z^2 \cup Y$ is a district in $\mathcal{H}[Z^1 \cup Z^2 \cup Y]$, which completes the proof.

$\square$

*Proof of Theorem 1.* A polynomial-time reduction from MCIP to the edge ID problem follows immediately from Proposition 3. MCIP is shown to be NP-hard [1]. As a result, the edge ID problem is Np-hard. □

## A.2 Reduction from edge ID to MCIP

**Proposition 2.** *There exists a polynomial-time reduction from edge ID to MCIP and vice versa.*

To prove Proposition 2, we begin with presenting a transformation $\mathcal{T}_2(\mathcal{G}, Y)$ which is in the core of reduction from edge ID to MCIP.

Suppose we want to solve the edge ID problem given ADMG $\mathcal{G} = (V^{\mathcal{G}}, E_d^{\mathcal{G}}, E_b^{\mathcal{G}})$, query $Q[Y]$, and edge weights $W_{\mathcal{G}} = \{w_e | e \in \mathcal{G}\}$. Let $X = V^{\mathcal{G}} \setminus Y$ denote the set of vertices of $\mathcal{G}$ excluding $Y$. We define the transformation $(\mathcal{H}, Y^{mcip}) = \mathcal{T}_2(\mathcal{G}, Y)$ where $\mathcal{H} = (V^{\mathcal{H}}, E_d^{\mathcal{H}}, E_b^{\mathcal{H}})$ is an ADMG and $Y^{mcip} \subseteq V^{\mathcal{H}}$ as follows. Note that $V^{\mathcal{H}}$ will consist of two disjoint set of vertices, namely $V_{top}^{\mathcal{H}}$ and $V_{bot}^{\mathcal{H}}$, i.e., $V^{\mathcal{H}} = V_{top}^{\mathcal{H}} \cup V_{bot}^{\mathcal{H}}$.

a. Begin with $V_{top}^{\mathcal{H}} = V_{bot}^{\mathcal{H}} = \emptyset, Y^{mcip} = \emptyset$. For any vertex $v \in V^{\mathcal{G}}$, add a vertex $v$ to $V_{top}^{\mathcal{H}}$ with cost $C(v) = \infty$. If $v \in Y$, add $v$ to $Y^{mcip}$.

b. For any directed edge $(v_i, v_j) \in E_d^{\mathcal{G}}$ with weight $w_{ij}^d$ in $\mathcal{G}$, add a new vertex $v_{ij}^d$ to $V_{top}^{\mathcal{H}}$, with cost $C(v_{ij}^d) = w_{ij}^d$, where

$$v_{ij}^d = \begin{cases} x_{ij}^d & \text{if } v_i, v_j \in X, \\ z_{ij}^d & \text{if } v_i \in Y \text{ or } v_j \in Y, \\ y_{ij}^d & \text{if both } v_i, v_j \in Y. \end{cases}$$

Draw directed edges $(v_i, v_{ij}^d)$ and $(v_{ij}^d, v_j)$. Further, draw a bidirected edge between $v_i$ and $v_{ij}^d$.

c. For any bidirected edge $\{x_i, x_j\} \in E_b^{\mathcal{G}}$ with weight $w_{ij}^b$, add a new vertex, $x_{ij}^b$ to $V_{top}^{\mathcal{H}}$ with cost $C(x_{ij}^b) = w_{ij}^b$. Add two bidirected edges $\{x_i, x_{ij}^b\}$ and $\{x_j, x_{ij}^b\}$. Further, draw two directed edges $(x_{ij}^b, x_i)$ and $(x_{ij}^b, x_j)$ in $\mathcal{H}$.

d. For any bidirected edge $\{x_i, y_j\}$ with weight $w_{ij}^b$, add a new vertex $z_{ij}^b$ to $V_{top}^{\mathcal{H}}$ with cost $C(z_{ij}^b) = w_{ij}^b$. Draw bidirected edges $\{z_{ij}^b, x_i\}$ and $\{z_{ij}^b, y_j\}$. Then draw a directed edge from $z_{ij}^b$ to $x_i$.

e. For any bidirected edge between $\{y_i, y_j\} \in E_b^{\mathcal{G}}$ with weight $w_{ij}^b$ in $\mathcal{G}$, add a new vertex, $y_{ij}^b$ to $V_{top}^{\mathcal{H}}$ with cost $C(y_{ij}^b) = w_{ij}^b$. Draw bidirected edges $\{y_{ij}^b, y_i\}$ and $\{y_{ij}^b, y_j\}$. Further, for any $x \in X$, draw a directed edge from $y_{ij}^b$ to $x$.

f. Let $y_1 \prec \ldots \prec y_k$ denote a topological ordering among vertices of $Y$. For every pair $\{y_i, y_j\}$ of vertices of $Y$, where $i < j$, add vertices $y_i^{ij}, y_{i+1}^{ij}, \ldots, y_j^{ij}$ to $V_{bot}^{\mathcal{H}}$. Add $y_j^{ij}$ to $Y^{mcip}$. Draw the directed edges $(y_k, y_k^{ij})$ for every $i \leq k \leq j$. Draw the directed edges $(y_k^{ij}, y_i^{ij})$ for every $i < k < j$, and the directed edge $(y_i^{ij}, y_i^{ij})$. Draw a bidirected edge between $y_j$ and $y_i^{ij}$. Further, for any bidirected edge $\{y_k, y_l\} \in E_b^{\mathcal{G}}$ where $i \leq k, l \leq j$, add a new vertex $y_{kl}^{ij}$ to $V_{bot}^{\mathcal{H}}$, draw two bidirected edges $\{y_{kl}^{ij}, y_k^{ij}\}$ and $\{y_{kl}^{ij}, y_l^{ij}\}$, and a directed edge $(y_{kl}^{ij}, y_{ij}^b)$. The costs of the all of the vertices in $V_{bot}^{\mathcal{H}}$ are infinite.

With abuse of notation, for any bidirected edge $e_{ij}^b = \{v_i, v_j\} \in E_b^{\mathcal{G}}$ and any directed edge $e_{ij}^d = (v_i, v_j) \in E_d^{\mathcal{G}}$, we define $\mathcal{T}_2(e_{ij}^b) = v_{ij}^b$ and $\mathcal{T}_2(e_{ij}^d) = v_{ij}^d$, respectively, where $v_{ij}^b, v_{ij}^d \in V^{\mathcal{H}}$ are the vertices representing their corresponding edges.

We will utilize the following results to prove Proposition 2. More precisely, Lemmas 2 through 9 are used to prove Proposition 4, which in turn is used to prove Proposition 2.

**Lemma 2.** *Suppose $\mathcal{G}$ is an ADMG, $Y$ is a set of its vertices, and $(\mathcal{H}, Y^{mcip}) = \mathcal{T}(\mathcal{G}, Y)$. Each vertex $y \in Y^{mcip}$ is a district in $\mathcal{H}$.*

*Proof.* It suffices to show that for every pair of $v_1, v_2 \in Y^{mcip}$ there is no bidirected edge between them in $\mathcal{H}$. Suppose first that $v_1, v_2 \in Y$. Any bidirected edge between $v_1$ and $v_2$ in $\mathcal{G}$ (if it exists)

607  is removed in step (e) of the transformation, and none of the steps (a) through (f) add a bidirected
608  edge between them. Otherwise, at least one of $v_1, v_2$, w.l.o.g. $v_1$, is in $Y^{mcip} \setminus Y$. Suppose w.l.o.g.
609  that $v_1 = y_j^{ij}$. From step (f) of the transformation $\mathcal{T}$, we know that $v_1$ has bidirected edges only to
610  vertices $y_{kj}^{ij}$, where none of them is a member of $Y^{mcip}$.  □

**Lemma 3.** *Suppose $\mathcal{G}$ is an ADMG, $Y$ is a set of its vertices, and $(\mathcal{H}, Y^{mcip}) = \mathcal{T}_2(\mathcal{G}, Y)$. Suppose*
612  *there is a hedge formed for $Q[y]$ in $\mathcal{H}$, where $y \in Y$. Let $H$ denote the set of vertices of this hedge. $H$*
613  *does not include any of the vertices added in the step (f) of the transformation. That is, $H \cap V_{bot}^{\mathcal{H}} = \emptyset$.*

*Proof.* Define $V_1 = \{y_{kl}^{ij} \in V_{bot}^{\mathcal{H}}, \forall i, j, k, l\}$, and $V_2 = V_{bot}^{\mathcal{H}} \setminus V_1$. By construction of $\mathcal{H}$, the vertices
615  of $V_2$ have directed edges only to vertices in $V_2$. Therefore, for each vertex $v \in V_2$, we have
616  $v \notin Anc_{\mathcal{H}[H]}(y)$. As a result, $V_2 \cap H = \emptyset$, since by definition of hedge, any vertex of $H$ is an
617  ancestor of $y$ in $\mathcal{H}[H]$. Now, consider an arbitrary vertex $v \in V_1$. By construction of $\mathcal{H}$, if there
618  exists a bidirected edge $\{v, v'\} \in E_b^{\mathcal{H}}$, we must have that $v' \in V_2$. Therefore, if $v \in H$, there must
619  be at least one vertex $v' \in V_2 \cap H$. Since we proved $V_2 \cap H = \emptyset$, $v$ cannot be in $H$. Consequently,
620  $V_1 \cap H = \emptyset$.

□

**Lemma 4.** *Suppose $\mathcal{G}$ is an ADMG, $Y$ is a set of its vertices, and $(\mathcal{H}, Y^{mcip}) = \mathcal{T}(\mathcal{G}, Y)$. Suppose*
623  *there is a hedge formed for $Q[y_j^{ij}]$ in $\mathcal{H}$, where $y_i, y_j \in Y$ and $y_j^{ij}$ is the vertex corresponding to the*
624  *pair $(y_i, y_j)$ added in step (f) of the transform $\mathcal{T}$. Let $H$ denote the set of vertices of this hedge. If*
625  $v \in H \cap V_{bot}^{\mathcal{H}}$, *then $v$ has the superscript $ij$, that is, $v$ is either one of the vertices $y_k^{ij}$, or one of the*
626  *vertices $y_{kl}^{ij}$, where $i \le k, l \le j$. In the latter case, $y_{kl}^b \in H$.*

*Proof.* Define $V_1 = \{y_{kl}^{mn} \in V_{bot}^{\mathcal{H}}, \forall m, n, k, l\}$, and $V_2 = V_{bot}^{\mathcal{H}} \setminus V_1$. Suppose $V_1^* = \{v_{kl}^{ij} \in$
628  $V_{bot}^{\mathcal{H}}, \forall k, l\}$ and $V_2^* = \{v_k^{ij} \in V_{bot}^{\mathcal{H}}, \forall k\}$. Also define $V_1' = V_1 \setminus V_1^*$, $V_2' = V_2 \setminus V_2^*$. For the first
629  part of the claim, it suffices to show that $V_1' \cap H = \emptyset, V_2' \cap H = \emptyset$. By construction of $\mathcal{H}$, the
630  vertices of $V_2'$ do not have any child out of $V_2'$. Therefore, $V_2' \cap Anc_{\mathcal{H}[H]}(y_j^{ij}) = \emptyset$. This implies that
631  $V_2' \cap H = \emptyset$. Now let $v_1^{i'j'}$ be an arbitrary vertex in $V_1'$. By construction of $\mathcal{H}$, $v_1^{i'j'}$ has bidirected
632  edges only to vertices of $V_2'$. This implies that if $v_1^{i'j'} \in H$, there must be at least one vertex of $V_2'$
633  in $H$ which is in contradiction with $V_2' \cap H = \emptyset$. Therefore, $v_1^{i'j'} \notin H$. Since $v_1^{i'j'}$ is an arbitrary
634  vertex in $V_1'$, we conclude $V_1' \cap H = \emptyset$.

Now, we prove that if $v \in H$ is one of the vertices $y_{kl}^{ij}$, we have $y_{kl}^b \in H$. Since $y_{kl}^{ij} \in H$, there exists
636  a directed path from $y_{kl}^{ij}$ to $y_j^{ij}$ in $\mathcal{H}[H]$. Since $y_{kl}^b$ is the only child of $y_{kl}^{ij}$, the aforementioned path
637  passes through $y_{kl}^b$. Therefore, $y_{kl}^b \in H$.

□

**Lemma 5.** *Suppose $\mathcal{G}' = (V^{\mathcal{G}'}, E_d^{\mathcal{G}'}, E_b^{\mathcal{G}'})$ is an ADMG, $Y \subseteq V^{\mathcal{G}'}$ is a set of its vertices, and*
640  $(\mathcal{H}', Y^{mcip}) = \mathcal{T}(\mathcal{G}', Y)$. *Let $E_d'' \subseteq E_d^{\mathcal{G}'}$ and $E_b'' \subseteq E_b^{\mathcal{G}'}$ be arbitrary edges of $\mathcal{G}$, and define*
641  $E_d^{\mathcal{G}} = E_d^{\mathcal{G}'} \setminus E_d'', E_b^{\mathcal{G}} = E_b^{\mathcal{G}'} \setminus E_b''$. *Define $\mathcal{G} = (V^{\mathcal{G}}, E_d^{\mathcal{G}}, E_b^{\mathcal{G}})$ and $\mathcal{H} = \mathcal{H}'[V^{\mathcal{H}'} \setminus V']$, where*
642  $V^{\mathcal{G}} = V^{\mathcal{G}'}$ *and $V' = \{v \in V^{\mathcal{H}'} | \exists e \in E_b'' \cup E_d'', v = \mathcal{T}_2(e)\}$. Suppose there is a hedge formed*
643  *for $Q[y_j^{ij}]$ in $\mathcal{H}$ for some $i, j$. Let $H$ denote the set of vertices of this hedge in $\mathcal{H}$. The set of*
644  *vertices $Y^* = \{y_k | y_k^{ij} \in H\}$ is a district in $\mathcal{G}[Y]$. Moreover, $H_{top} = Anc_{\mathcal{H}[H_{top}]}(Y^*)$, where*
645  $H_{top} = H \cap V_{top}^{\mathcal{H}}$.

*Proof.* First we prove that $Y^*$ is a district in $\mathcal{G}[Y]$. Consider an arbitrary vertex $y_k^{ij}$ in $H$. By definition
647  of hedge, there exists a bidirected path, $p_1$, between $y_k^{ij}$ and $y_j^{ij}$ in $\mathcal{H}[H]$. Let $Y^{ij}$ denotes the set of
648  vertices in $H$ such that their superscript is $ij$. Lemma 4 implies that $H \subseteq V_{top}^{\mathcal{H}} \cup Y^{ij}$. Furthermore,
649  by construction of $\mathcal{H}$, there is only one bidirected edge between $Y^{ij}$ and $H \setminus Y^{ij}$, which is $\{y_j, y_i^{ij}\}$.
650  Therefore, all of the vertices on the path $p_1$ are in $Y^{ij}$. Now, we define $Y_1' = \{y_k | y_k^{ij} \in p_1\}$ and

$Y_2' = \{y_{kl}^b | y_{kl}^{ij} \in p_1\}$, i.e., the $V_{top}^{\mathcal{H}}$ counterparts of the vertices in $p_1$. Since the vertices on $p_1$ are in $H$, $Y_1' \subseteq Y^*$. From Lemma 4, we know that if $y_{kl}^{ij} \in H$, then, $y_{kl}^b \in H$. It implies that $Y_2' \subseteq H$. As a result, $Y_1'$ and $Y_2'$ are both vertices of $\mathcal{H}$. Now if we replace all the vertices in $p_1$ with their corresponding counterpart in $Y_1' \cup Y_2'$, we arrive at a bidirected path $p_2$ between $y_k$ and $y_j$ in $\mathcal{H}[Y_1' \cup Y_2']$ (as by construction the same edges exist in $V_{top}^{\mathcal{H}}$). By definition of $\mathcal{G}$ and $\mathcal{H}$, if a vertex $y_{kl}^b$ exists in $\mathcal{H}$, the corresponding edge $\{y_k, y_l\}$ exists in $\mathcal{G}$. As a result, a bidirected path between $y_k$ and $y_l$ exists in $\mathcal{G}[Y_1']$. Noting that $y_k$ is an arbitrary vertex in $Y^*$ and $Y_1' \subseteq Y^*$, this implies that all of the vertices of $Y^*$ are in the same district as $y_j$ in $\mathcal{G}[Y^*]$, which completes the proof.

Next, we prove that $H_{top} = Anc_{\mathcal{H}[H_{top}]}(Y^*)$. To this end, it suffices to show that there is a directed path form an arbitrary vertex $v \in H_{top}$ to $Y^*$ in $\mathcal{H}[H_{top}]$. Since $H$ forms a hedge for $Q[y_j^{ij}]$ in $\mathcal{H}$, there exists a directed path from $v$ to $y_j^{ij}$ in $\mathcal{H}[H]$. This path must go through the only parent of $y_j^{ij}$, which is $y_i^{ij}$. Then, the last vertex on the path is one of the parents of $y_i^{ij}$. If this parent is $y_i$, we are done as we have a directed path from $v$ to $y_i$, where $y_i \in Y^*$ and it has no ancestors in $H \setminus H_{top}$. Otherwise, let this parent be $y_k^{ij}$ for some $i < k < j$. Now the last vertex on the path before $y_k^{ij}$ must be $y_k$, which is the only parent of $y_k^{ij}$. Note that by definition of $Y^*$, $y_k \in Y^*$. Therefore, $v$ has a directed path to $Y^*$ in $\mathcal{H}[H_{top}]$. $\square$

**Lemma 6.** *Suppose $\mathcal{G} = (V^\mathcal{G}, E_d^\mathcal{G}, E_b^\mathcal{G})$ is an ADMG, $Y$ is a set of its vertices, and $(\mathcal{H}, Y^{mcip}) = \mathcal{T}_2(\mathcal{G}, Y)$. Suppose there is a hedge formed for $Q[y]$ in $\mathcal{H}$ for some $y \in Y^{mcip}$. Let $H$ denote the set of vertices of this hedge. Then $H \cap X \neq \emptyset$, where $X = V^\mathcal{G} \setminus Y$.*

*Proof.* Since $H$ forms a hedge for $Q[y]$ in $\mathcal{H}$, there exists a vertex $h \in H$ such that $\{y, h\} \in E_b^\mathcal{H}$. There are two possibilities for $y \in Y^{mcip}$:

- $y = y_i \in Y$. From Lemma 4 we have $h \notin V_{bot}^\mathcal{H}$. Therefore, by construction of $\mathcal{H}$, $h = y_{ij}^b$ for some $j$.

- $y = y_j^{ij} \in V_{bot}^\mathcal{H}$. By construction of $\mathcal{H}$, $h = y_{kj}^{ij}$ for some $k$. Vertex $h$ must have a directed path to $y$ in $H$ by definition of hedge, which must go through the only child of $h$, i.e., $y_{kl}^b$.

In both cases, we showed that there exists a vertex $v = y_{ij}^b \in H$ for some $i, j$. By definition of hedge, there is a bidirected path, $p$, from $v$ to $y$ in $\mathcal{H}$ because $v \in Anc_\mathcal{H}(y)$. Since all of the children of $v$ are in $X$, there is at least one vertex in $X$ on path $p$. Therefore, $H$ includes at least one vertex of $X$.

$\square$

**Lemma 7.** *[Inverse transform preserves hedges.] Suppose $\mathcal{G}' = (V^{\mathcal{G}'}, E_d^{\mathcal{G}'}, E_b^{\mathcal{G}'})$ is an ADMG, $Y \subseteq V^{\mathcal{G}'}$ is a set of its vertices, and $(\mathcal{H}', Y^{mcip}) = \mathcal{T}_2(\mathcal{G}', Y)$. Let $E_d'' \subseteq E_d^{\mathcal{G}'}$ and $E_b'' \subseteq E_b^{\mathcal{G}'}$ be arbitrary edges of $\mathcal{G}$, and define $E_d^\mathcal{G} = E_d^{\mathcal{G}'} \setminus E_d''$, $E_b^\mathcal{G} = E_b^{\mathcal{G}'} \setminus E_b''$. Define $\mathcal{G} = (V^\mathcal{G}, E_d^\mathcal{G}, E_b^\mathcal{G})$ and $\mathcal{H} = \mathcal{H}'[V^{\mathcal{H}'} \setminus V']$, where $V^\mathcal{G} = V^{\mathcal{G}'}$ and $V' = \{v \in V^{\mathcal{H}'} | \exists e \in E_b'' \cup E_d'', v = \mathcal{T}_2(e)\}$. Let $W \subseteq V_{top}^\mathcal{H}$ be a set of vertices of $\mathcal{H}$. Let $W_s \subseteq W \cap V^\mathcal{G}$ be a subset of $W$ such that $W_s$ are vertices of $\mathcal{G}$ as well. Consider the inverse transform of $\mathcal{H}[W]$ in the ADMG $\mathcal{G}$, i.e., for any $v = v_{ij}^b \in W$, delete $v$ and all edges incident to it and draw a bidirected edge between $v_i$ and $v_j$, and for any $v = v_{ij}^d$, delete $v$ and all edges incident to it and draw a directed edge from $v_i$ to $v_j$. Let the resulting subgraph (which is a subgraph of $\mathcal{G}$) be denoted by $\mathcal{G}[W^{-1}]$ with the set of vertices $W^{-1} \subseteq V^\mathcal{G}$. If $Anc_{\mathcal{H}[W]}(W_s) = W$, then $Anc_{\mathcal{G}[W^{-1}]}(W_s) = W^{-1}$. Moreover, if $W$ is a district in $\mathcal{H}[W]$, then $W^{-1}$ is a district in $\mathcal{G}[W^{-1}]$.*

*Proof.* First, we show that if $Anc_{\mathcal{H}[W]}(W_s) = W$, then $Anc_{\mathcal{G}[W^{-1}]}(W_s) = W^{-1}$. Let $v$ be an arbitrary vertex in $W^{-1}$. Vertex $v$ is in $W$ because $W^{-1} \subseteq W$. Since $v \in W$ and $v \in Anc_{\mathcal{H}[W]}(W_s)$, $v$ has a directed path $v \to \ldots v_i \to v_{ij}^d \to v_j \cdots \to w$, denoted by $l$, to a vertex $w \in W_s$ in $\mathcal{H}[W]$. For each vertex $v_{ij}^d$ on path $l$, we have $v_i, v_j \in \mathcal{G}[W^{-1}]$ and since $v_{ij}^d \in V^\mathcal{H}$, by definition of $\mathcal{G}$ and $\mathcal{H}$, there exists $(v_i, v_j) \in E_d^\mathcal{G}$ s.t. $i \prec j$, and consequently, $(v_i, v_j) \in E_d^{\mathcal{G}[W^{-1}]}$. Therefore,

there exists a directed path from $v$ to $w$ in $\mathcal{G}[W^{-1}]$. Noting that $v$ is an arbitrary vertex in $W^{-1}$, we conclude that $Anc_{\mathcal{G}[W^{-1}]}(W_s) = W^{-1}$.

Now, we prove that if $W$ is a district in $\mathcal{H}[W]$, then $W^{-1}$ is a district in $\mathcal{G}[W^{-1}]$. Consider two vertices $v_1, v_2 \in W^{-1}$. Since $v_1, v_2 \in W$ and $W$ is a district, there exists a bidirected path $v_1 \leftrightarrow \ldots v_i \leftrightarrow v_{ij}^b \leftrightarrow v_j \cdots \leftrightarrow v_2$, denoted by $p$, between $v_1$ and $v_2$ in $\mathcal{H}[W]$. Each vertex $v_{ij}^b$ on path $p$ is in $\mathcal{H}$ and $v_i, v_j \in \mathcal{G}[W^{-1}]$. By definition of $\mathcal{G}$ and $\mathcal{H}$, we have $\{v_i, v_j\} \in E_b^{\mathcal{G}}$. Therefore, $\{v_i, v_j\} \in E_b^{\mathcal{G}[W^{-1}]}$. Then, there is a bidirected path between $v_1$ and $v_2$ in $\mathcal{G}[W^{-1}]$. Since $v_1$ and $v_2$ are two arbitrary vertices in $W^{-1}$, it implies that $W^{-1}$ is a district in $\mathcal{G}[W^{-1}]$. $\square$

**Lemma 8.** *[Transform preserves hedges.] Suppose $\mathcal{G}' = (V^{\mathcal{G}'}, E_d^{\mathcal{G}'}, E_b^{\mathcal{G}'})$ is an ADMG, $Y \subseteq V^{\mathcal{G}'}$ is a set of its vertices, and $(\mathcal{H}', Y^{mcip}) = \mathcal{T}_2(\mathcal{G}', Y)$. Let $E_d'' \subseteq E_d^{\mathcal{G}'}$ and $E_b'' \subseteq E_b^{\mathcal{G}'}$ be arbitrary edges of $\mathcal{G}$, and define $E_d^{\mathcal{G}} = E_d^{\mathcal{G}'} \setminus E_d''$, $E_b^{\mathcal{G}} = E_b^{\mathcal{G}'} \setminus E_b''$. Define $\mathcal{G} = (V^{\mathcal{G}}, E_d^{\mathcal{G}}, E_b^{\mathcal{G}})$ and $\mathcal{H} = \mathcal{H}'[V^{\mathcal{H}'} \setminus V']$, where $V^{\mathcal{G}} = V^{\mathcal{G}'}$ and $V' = \{v \in V^{\mathcal{H}'} | \exists e \in E_b'' \cup E_d'', v = \mathcal{T}_2(e)\}$. Let $W \subseteq V^{\mathcal{G}}$ be a set of vertices of $\mathcal{G}$ such that $W \setminus Y \neq \emptyset$. Let $W_s \subseteq W$ be a subset of $W$. Let the transformed graph of $\mathcal{G}[W]$ under $\mathcal{T}_2$ be denoted by $\mathcal{H}''$, where $\mathcal{H}'' \subseteq \mathcal{H}$. Define $W^* = V_{top}^{\mathcal{H}''}$. If $Anc_{\mathcal{G}[W]}(W_s) = W$, then $Anc_{\mathcal{H}[W^*]}(W_s) = W^*$. Moreover, if $W$ is a district in $\mathcal{G}[W]$, then $W^*$ is a district in $\mathcal{H}[W^*]$.*

*Proof.* First, we prove that if $Anc_{\mathcal{G}[W]}(W_s) = W$, then $Anc_{\mathcal{H}[W^*]}(W_s) = W^*$. Take an arbitrary vertex $v \in W^*$. There are two possibilities for $v$:

- $v \in W$. That is, vertex $v$ is in $\mathcal{G}[W]$.

- $v \notin W$. This implies that $v$ represents an edge $e$ between two vertices $v_i$ and $v_j$ in $\mathcal{G}[W]$. There are three possibilities for $e$:

    - $e = (v_i, v_j)$. By construction of $\mathcal{H}$, $v$ is parent of $v_j$ in $\mathcal{H}[W^*]$, where $v_j$ is a vertex of $\mathcal{G}[W]$.

    - $e = \{v_i, v_j\}$ and $v_i \in X$ or $v_j \in X$. In this case, $v$ is parent of at least one of $v_i$ and $v_j$ in $\mathcal{H}[W^*]$, w.l.o.g. $v_i$, where $v_i$ is a vertex of $\mathcal{G}[W]$.

    - $e = \{v_i, v_j\}$ and $v_i, v_j \in Y$. By construction of $\mathcal{H}$, $v$ is parent of all vertices in $V^{\mathcal{G}} \setminus Y$. Since $W \setminus Y \neq \emptyset$, there exists a vertex $x$ in $\mathcal{G}[W]$ such that $v$ is a parent of $x$.

    In all three cases above, we proved that there exists a vertex $x \in W$ such that $v$ is a parent of $x$.

Therefore, we showed that any vertex $v \in W^*$ either is itself a vertex in $W$ or is a parent of a vertex in $W$. As a result, it suffices to show that every $w \in W$ has a directed path to $W_s$ in $\mathcal{H}[W^*]$. We know that $w$ has a directed path to $W_s$ in $\mathcal{G}[W]$ such as $p$. Take an arbitrary pair of consecutive vertices on this path, such as $v_1$ and $v_2$. The directed edge $(v_1, v_2)$ exists in $\mathcal{G}[W]$. As a result, the directed path $v_1 \rightarrow v_{12}^d \rightarrow v_2$ exists in $\mathcal{H}[W^*]$. Starting at $w$ and repeating this argument for every pair of consecutive vertices on $p$, we conclude that there exists a directed path from $w$ to $W_s$, which completes the proof.

Now, we show that if $W$ is a district in $\mathcal{G}[W]$, then $W^*$ is a district in $\mathcal{H}[W^*]$. Take an arbitrary vertex $v \in W^*$. There are two possibilities for $v$:

- $v \in W$. That is, $v$ is a vertex in $\mathcal{G}[W]$.

- $v \notin W$. In this case, at least one of the vertices $v$ represents an edge $e$ between two vertices $v_i$ and $v_j$ in $\mathcal{G}[W]$. By construction of $\mathcal{H}$, $v$ is connected to at least one of $v_i$ or $v_j$, w.l.o.g. $v_i$, by a bidirected edge, where $v_i \in W$.

We showed that any vertex $v \in W^*$ either is in $W$, or is connected to a vertex in $W$ through a bidirected edge. Therefore, it suffices to show that for any two vertices $w_1, w_2 \in W$ there exists a bidirected path between $w_1$ and $w_2$ in $\mathcal{H}[W^*]$. Since $w_1, w_2 \in W$, there is a bidirected path, $p$, between $w_1$ and $w_2$ in $\mathcal{G}[W]$. Take an arbitrary pair of consecutive vertices on this path, such as $v_1$ and $v_2$. The bidirected edge $\{v_1, v_2\}$ exists in $\mathcal{G}[W]$. As a result, the bidirected path $v_1 \leftrightarrow v_{12}^b \leftrightarrow v_2$

742 exists in $\mathcal{H}[W^*]$. Starting at $w$ and repeating this argument for every pair of consecutive vertices on
743 $p$, we conclude that there exists a bidirected path from $w_1$ to $w_2$, which completes the proof. $\qquad\square$

**Lemma 9.** *Suppose $\mathcal{G}$ is an ADMG, and $Y$ is a subset of its vertices. Also let $Y^*$ be a district in*
745 $\mathcal{G}[Y]$. *If the set of vertices $H$ form a hedge for $Q[Y^*]$, then $H \setminus Y \neq \emptyset$.*

746 *Proof.* Assume by contradiction $H \setminus Y = \emptyset$, i.e., $H \subseteq Y$. By definition of hedge, we know
747 $H \setminus Y^* \neq \emptyset$. Take an arbitrary vertex $v \in H \setminus Y^*$. Furthermore, $v \in Y \setminus Y^*$ because $H \subseteq Y$. Since
748 $H$ forms a hedge for $Q[Y^*]$, $H$ is a district in $\mathcal{G}[H]$. Therefore, there exists a bidirected path between
749 $v$ and a vertex $y^* \in Y^*$ in $Q[Y]$ which is in contradiction with the assumption that $Y^*$ is a district in
750 $\mathcal{G}[Y]$. $\qquad\square$

**Proposition 4.** *Suppose $\mathcal{G}' = (V^{\mathcal{G}'}, E_d^{\mathcal{G}'}, E_b^{\mathcal{G}'})$ is an ADMG, $Y \subseteq V^{\mathcal{G}'}$ is a set of its vertices, and*
752 $(\mathcal{H}', Y^{mcip}) = \mathcal{T}_2(\mathcal{G}', Y)$. *Let $E_d'' \subseteq E_d^{\mathcal{G}'}$ and $E_b'' \subseteq E_b^{\mathcal{G}'}$ be arbitrary edges of $\mathcal{G}$, and define*
753 $E_d^{\mathcal{G}} = E_d^{\mathcal{G}'} \setminus E_d''$, $E_b^{\mathcal{G}} = E_b^{\mathcal{G}'} \setminus E_b''$. $Q[Y]$ *is identifiable in $\mathcal{G} = (V^{\mathcal{G}}, E_d^{\mathcal{G}}, E_b^{\mathcal{G}})$ if and only if $Q[Y^{mcip}]$*
754 *is identifiable in $\mathcal{H} = \mathcal{H}'[V^{\mathcal{H}'} \setminus V']$, where $V^{\mathcal{G}} = V^{\mathcal{G}'}$ and $V' = \{v \in V^{\mathcal{H}'} | \exists e \in E_b'' \cup E_d'', v =$*
755 $\mathcal{T}_2(e)\}$.

756 *Proof.* We prove the contrapositive, i.e., $Q[Y]$ is not identifiable in $\mathcal{G}$ iff $Q[Y^{mcip}]$ is not identifiable
757 in $\mathcal{H}$.

758 *If part.* Suppose $Q[Y^{mcip}]$ is not identifiable in $\mathcal{H}$. That is, there exists a hedge formed for $Q[Y^{mcip}]$
759 in $\mathcal{H}$. From Lemma 2, this hedge is formed for $Q[y']$ for some $y' \in Y^{mcip}$. Denote the set of vertices
760 of this hedge by $H$. We consider two possibilities separately:

- $y' = y_i$, where $y_i \in Y$. From Lemma 3, $H \subseteq V_{top}^{\mathcal{H}}$. Taking $W = H$ in Lemma 7, $W^{-1}$ is a
762 set of vertices in $\mathcal{G}$ such that $Anc_{\mathcal{G}[W^{-1}]}(y) = W^{-1}$, and $W^{-1}$ is a district in $\mathcal{G}$. Now take
763 $Y^*$ to be the district of $\mathcal{G}[Y]$ that includes $y_i$. By definition of hedge, $\mathcal{G}[W^{-1} \cup Y^*]$ forms a
764 hedge for $Q[Y^*]$ in $\mathcal{G}$. Note that from Lemma 6, $W^{-1} \setminus Y \neq \emptyset$. As a result, $Q[Y]$ is not
765 identifiable in $\mathcal{G}$.

- $y' = y_j^{ij}$, where $y_i, y_j \in Y$ and $y'$ is one of the vertices added to $\mathcal{H}$ in the last step of the
767 transformation $\mathcal{T}$ (step (f)). Define the set $Y^* = \{y_k | y_k^{ij} \in H\}$. From Lemma 5, $Y^*$ is a
768 district in $\mathcal{G}$, and therefore a district in $\mathcal{G}[Y]$. As a result, it suffices to show that there exists
769 a hedge formed for $Q[Y^*]$ in $\mathcal{G}$. Now define $H_{top} = H \cap V_{top}^{\mathcal{H}}$. By definition of hedge,
770 $H$ is a district in $\mathcal{H}[H]$, i.e., it is connected over its bidirected edges. By construction of
771 $\mathcal{H}$, there is only one bidirected edge between the vertices in $H_{top}$ and $H \setminus H_{top}$, which is
772 the bidirected edge between $y_j$ and $y_i^{ij}$. Therefore, this edge is a cut set that partitions the
773 graph $\mathcal{H}[H]$ into two connected components $\mathcal{H}[H_{top}]$ and $\mathcal{H}[H \setminus H_{top}]$. That is, $\mathcal{H}[H_{top}]$
774 is connected over its bidirected edges and therefore $H_{top}$ is a district in $\mathcal{H}[H_{top}]$. Further,
775 from Lemma 5, $H_{top} = Anc_{\mathcal{H}[H_{top}]}(Y^*)$. Noting that $H_{top} \subseteq V_{top}^{\mathcal{H}}$, taking $W = H_{top}$ in
776 Lemma 7, $W^{-1}$ is a district in $\mathcal{G}$ and $Anc_{\mathcal{G}[W^{-1}]}(Y^*) = W^{-1}$. Note that from Lemma 6,
777 $W^{-1} \setminus Y \neq \emptyset$. Therefore, the set of vertices $W^{-1}$ form a hedge for $Q[Y^*]$ in $\mathcal{G}$. Hence,
778 $Q[Y]$ is not identifiable in $\mathcal{G}$.

779 *Only if part.* Suppose $Q[Y]$ is not identifiable in $\mathcal{G}$. It implies that there exists a district of $\mathcal{G}[Y]$ such
780 as $Y^*$ such that there is a hedge formed for $Q[Y^*]$ in $\mathcal{G}$. Let $H$ denote the set of vertices of this hedge.
781 From Lemma 9, $H \setminus Y \neq \emptyset$. Define $W^*$ as in Lemma 8, that is the transform $\mathcal{T}(\mathcal{G}[H], Y^*)$ without
782 step (f) (only on the vertices of $V_{top}^{\mathcal{H}}$). Note that $Y^* \subseteq W^*$. We consider the following two cases
783 separately:

- $Y^* = \{y\}$, that is, $Y^*$ is a single vertex. From Lemma 8, $W^*$ is a district in $\mathcal{H}[W^*]$, and
785 $Anc_{\mathcal{H}[W^*]}(y) = W^*$. By definition of hedge, the vertices $W^*$ form a hedge for $Q[y]$ in $\mathcal{H}$.
786 Note that $y \in Y^{mcip}$, and from Lemma 2 it is a district of $\mathcal{H}[Y^{mcip}]$. As a result, $Q[Y^{mcip}]$
787 is not identifiable in $\mathcal{H}$.

- $|Y^*| \geq 2$. Let $y_i$ and $y_j$ be the first and the last vertices of $Y^*$ in the topological order. Define $Y^{ij*} = \{y_k^{ij}|y_k \in Y^*\} \cup \{y_{kl}^{ij}|y_k, y_l \in Y^*\}$. $Y^{ij*}$ are the vertices in $V_{bot}^{\mathcal{H}}$ with superscript $ij$ corresponding to the vertices in $Y^*$. Note that $y_i^{ij}, y_j^{ij} \in Y^{ij*}$, since $y_i, y_j \in Y^*$. Since $y_j^{ij} \in Y^{mcip}$ and from Lemma 2 $y_j^{ij}$ is a district in $\mathcal{H}[Y^{mcip}]$, it suffices to show that there is a hedge formed for $y_j^{ij}$ in $\mathcal{H}$. We show that the vertices $W = W^* \cup Y^{ij*}$ form a hedge for $y_j^{ij}$ in $\mathcal{H}$. From Lemma 8, $Anc_{\mathcal{H}[W^*]}(Y^*) = W^*$, that is, all of the vertices in $W^*$ are ancestors of $Y^*$ in $\mathcal{H}[W^*]$, and therefore in $\mathcal{H}[W]$. Also, the vertices $y_{kl}^{ij}$ in $Y^{ij*}$ have a direct edge to their corresponding vertex in $W^*$, i.e., $y_{kl}^b$, and therefore are ancestors of $Y^*$ in $\mathcal{H}[W]$ as well. Further, each vertex in $Y^*$ such as $y_k$ is a parent of $y_k^{ij}$, which is in turn a parent of $y_i^{ij}$ (or is $y_i^{ij}$ itself if $k = i$.) Finally, $y_i^{ij}$ has a directed edge to $y_j^{ij}$ by construction. As a result, all of the vertices $W$ have a direct path to $y_j^{ij}$ in $\mathcal{H}[W]$. That is, $Anc_{\mathcal{H}[W]}(y_j^{ij}) = W$. It now remains to show that $W$ is a district in $\mathcal{H}[W]$. From Lemma 8, $W^*$ is a district in $\mathcal{H}[W^*]$. As a result, the vertices $W^*$ are connected through bidirected edges in $\mathcal{H}[W]$. There is a bidirected edge between $y_j$ and $y_i^{ij}$ by construction of $\mathcal{H}$. It suffices to show that for any $v \in Y^{ij*}$, there exists a bidirected path between $v$ and $y_i^{ij}$ in $\mathcal{H}[W]$. A vertex $y_{kl}^{ij} \in Y^{ij*}$ (with double subscript, which are due to the bidirected edges among $Y^*$) has bidirected edges to $y_k^{ij}$ and $y_l^{ij}$, which are both in $Y^{ij*}$ by definition. Now take an arbitrary vertex $y_k^{ij} \in Y^{ij*}$ (with single subscript, due to vertices in $Y^*$). We know $y_k \in Y^*$, as $y_k^{ij} \in Y^{ij*}$, by definition of $Y^{ij*}$. $Y^*$ is a district in $\mathcal{G}[Y^*]$. That is, there exists a bidirected path from $y_k$ to $y_i$ in $\mathcal{G}[Y^*]$. From Lemma 8 by taking $W = Y^*$, there is a bidirected path $p$ from $y_k$ to $y_i$ in $\mathcal{H}[Y^* \cup \{y_{lm}|y_l, y_m \in Y^*\}]$. By construction of $\mathcal{H}$, if we replace each vertex $v$ on $p$ by $v^{ij}$, we achieve a bidirected path $p'$ with vertices in $Y^{ij*}$ from $y_k^{ij}$ to $y_i^{ij}$, which completes the proof.

$\square$

*Proof of Proposition 2.* The reduction from the edge ID problem to MCIP was shown through the proof of Proposition 4. The opposite direction is an immediate corollary of Proposition 3. $\square$

**Corollary 2.** *The edge ID problem and MCIP are equivalent.*

# B Maximal Hedge

---
**Algorithm 3** Maximal Hedge.

---
1: **function** MH($\mathcal{G}, Y$)
2:     Initialize $M \leftarrow \emptyset$
3:     **for** $Y_i$ in districts of $\mathcal{G}[Y]$ **do**
4:         $M \leftarrow M \cup$ **HHull**($\mathcal{G}, Y_i$)
5:     **return** $\mathcal{G}[M]$

---
1: **function** HHULL($\mathcal{G}, Y_i$)
2:     Initialize $H \leftarrow V^{\mathcal{G}}$
3:     **while** True **do**
4:         $C \leftarrow$ connected component (district) of $Y_i$ via bidirected edges in $\mathcal{G}[H]$
5:         $A \leftarrow$ ancestors of $Y_i$ in $\mathcal{G}[C]$
6:         **if** $C \neq A$ **then**
7:             $H \leftarrow A$
8:         **else**
9:             **break**
10:     **return** $H$

---

Herein, we present the algorithm for recovering the maximal hedge formed for $Q[Y]$ in a given ADMG $\mathcal{G}$ (see Definition 5). Maximal hedge was initially defined in [1] under the name *hedge hull*.

$$z \xrightarrow{\quad p \quad} x \xrightarrow{\quad q \quad} y$$

Figure 6: An example where the expert is aware that there is no causal path from $z$ to $y$, e.g., because $z \perp\!\!\!\perp y$ with high confidence.

We adopt the same definition, and when $\mathcal{G}[Y]$ comprises several districts, we define the maximal hedge as the union of the hedge hulls formed for each district of $\mathcal{G}[Y]$. As a result, the complete procedure of recovering the maximal hedge for a query $Q[Y]$, summarized in Algorithm 3, finds the maximal hedge formed for each district $Y_i$ of $\mathcal{G}[Y]$ and returns the union of them. This procedure is used as a subroutine **MH** in Algorithm 1. The function **HHull** is in fact Algorithm 1 borrowed from [1]. This function is proven to recover the union of all hedges formed for $Y_i$, where $Y_i$ is one of the districts of $\mathcal{G}[Y]$ (see Lemma 6 of [1]).

# C    Generalizing Assumption 1

Lemma 1 states the equivalence of Problems 1 and 2 with the edge ID problem under Assumption 1. However, as mentioned in the main text, this equivalence holds in the more general setting where we allow for perfect negative correlations among edges. As an example, consider the graph of Figure 6. Suppose that the performed statistical independence tests show that the two variables $z$ and $y$ are independent of each other with high confidence. As a result, the expert believes that the edges $(z, x)$ and $(x, y)$ must not exist simultaneously, as otherwise the causal path from $z$ to $y$ would make them dependent. In such cases, the belief of the expert can be modeled as probabilities $p$ and $q$ assigned to the existence of the edges $(z, x)$ and $(x, y)$, respectively, as well as a perfect negative correlation between them.

Note that the aforementioned constraint, i.e., that the edges do not exist simultaneously, can be specified for any number of edges, not limited to two edges only. For instance, the expert might believe at least one of the edges along a causal path of length $n$ must not exist in the true ADMG describing the system. This belief can be modeled as an extra constraint in the optimization of Equations 2 and 3. We show that with the specification of such negative correlations, Problems 1 and 2 can still be cast as an instance of the edge ID problem. Therefore, the results presented in this work are valid in this setting as well.

**Assumption 2.** *The edges in $\mathcal{G}$ are assigned probabilities $p_e, \forall e \in \mathcal{G}$, and perfect negative correlations are defined among subsets of edges. More precisely, for any subset $E \subseteq E_d^{\mathcal{G}} \cup E_b^{\mathcal{G}}$, there is either 1) no constraint (mutually independent), or 2) the constraint that at least one of the edges in $E$ must not exist in the true ADMG (perfect negative correlation).*

**Proposition 5.** *Under Assumption 2, there exists a reduction from Problems 1 and 2 to the edge ID problem and vice versa with the time complexity in the order of $\mathcal{O}(|C| \cdot |V^{\mathcal{G}}| + |E_d^{\mathcal{G}} \cup E_b^{\mathcal{G}}|)$, where $C$ is the set of perfect correlation constraints.*

*Proof.* First note that we proved the equivalence of Problems 1 and 2 with the edge ID problem without the perfect correlation constraints in Lemma 1. As a result, under assumption 2, i.e., by adding the perfect correlation constraints, Problems 1 and 2 are equivalent to a modified edge ID problem with those constraints. But we claim that there exists and instance of the original unconstrained edge ID problem which is equivalent to these problems. To see this, first note that we know from Corollary 2 that the edge ID problem is equivalent to MCIP. Therefore, it suffices to show that there exists an instance of MCIP which is equivalent to the constrained edge ID mentioned above. To this end, consider the transform $\mathcal{T}_2(\mathcal{G}, Y)$ introduced in Section A.2. This transformation maps an instance of the edge ID problem to an instance of MCIP. Applying this transformation to the constrained edge ID problem, we can map the constrained edge ID to an instance of MCIP with extra constraints, with transforming the constraints as well. That is, if for instance, there is a perfect negative correlation among the edges $e_1, e_2$ in $\mathcal{G}$, this constraint is mapped to a negative perfect correlation on the corresponding vertices in $\mathcal{H}$, namely $\mathcal{T}_2(e_1), \mathcal{T}_2(e_2)$. In words, this constraint would be that at least one of $\mathcal{T}_2(e_1)$ and $\mathcal{T}_2(e_2)$ must be intervened upon. We show that such constraints can be integrated into the original definition of MCIP.

Suppose we have an MCIP problem in ADMG $\mathcal{G}$ with query $Q[Y]$, with the extra constraint that at least one of the vertices $X \subseteq V^{\mathcal{G}}$ must be intervened upon. Consider the example of $X =$

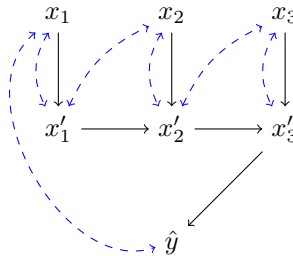

Figure 7: Integrating the perfect negative correlation constraint into MCIP.

$\{x_1, x_2, x_3\}$ in Figure 7. We build a new ADMG $\mathcal{G}'$ by adding vertices $\{x'|x \in X\}$, i.e., a new vertex corresponding to each vertex in $X$, along with an auxiliary vertex $\hat{y}$ to $\mathcal{G}$. We fix a random ordering over the vertices of $X$, and denote the set of vertices of $X$ as $x_1, ..., x_m$. We add the directed edges $(x_i, x'_i)$ to $\mathcal{G}'$, as well as the bidirected edges $\{x_i, x'_i\}$. Further, we draw directed edges $(x'_i, x'_{i+1})$ for every $1 \leq i < m$. Finally, we draw the directed edge $(x'_m, \hat{y})$ and the bidirected edge $\{x_1, \hat{y}\}$. Refer to the graph in Figure 7 for an example with $X = \{x_1, x_2, x_3\}$. Note that the set $X \cup X' \cup \{\hat{y}\}$ forms a hedge for $Q[\hat{y}]$, where $X' = \{x'|x \in X\}$ Now it suffices to set the cost of intervention on vertices of $X'$ to infinity, and consider MCIP for the query $Q[Y \cup \{\hat{y}\}]$ in $\mathcal{G}'$. It is straightforward to see that the objective of this problem would be to find the minimum cost intervention for identification of $Q[Y]$, with the constraint that at least one of the vertices in $X$ must be intervened on. Note that as soon as one vertex in $X$ gets intervened upon, there is no hedge left for $Q[\hat{y}]$. Also it is noteworthy that adding this structure does not add any new hedges formed for $Q[Y]$, since the structure only includes new descendants for $X$ which have no directed paths to $Y$. Also note that the vertices $X'$ and $\hat{y}$ are specific to the very constraint corresponding to the set of vertices $X$. For any constraint, we add such a structure to $\mathcal{G}$. The number of vertices (and therefore the time complexity) is at most in the order $\mathcal{O}(|C| \cdot |V^{\mathcal{G}}|)$, where $C$ is the set of constraints.

$\square$

## C.1 Further applications

The relaxation provided in this Appendix allows the approaches proposed in this work to be applicable to a more general set of problems. Herein, we discuss one such application.

Let us assume we run our algorithm which returns the subgraph with the highest probability, $\mathcal{G}_1$. However, the probability that $\mathcal{G}_1$ is the true causal structure describing the system might be too low. In such a case, the researcher might be interested in having a ranking of most probable graphs (for instance, the 10 most probable graphs), rather than only one most probable graph. This could be helpful for instance, when a unique identification formula is valid in a few of these graphs, or the researcher is interested in identifying more than one causal query. The methods discussed in this work along with the relaxation proposed in this appendix provide the tools to recover such a ranking (of the most probable graphs in which a query is identifiable). To see this, note that based on what we proposed in this Appendix, perfect negative correlation constraints can be added to the edge ID problem without additional computational cost. So we begin by solving the original problem, which yields a graph $\mathcal{G}_1$. We then solve it for a second time (i.e., re-run the algorithm), with the only difference that we add the perfect negative correlation constraint over the set of all edges of $\mathcal{G}_1$ (i.e., we force the algorithm to exclude at least one of the edges of $\mathcal{G}_1$.) The solution to this problem (let us call it $\mathcal{G}_2$) is the highest probability graph among all subgraphs except $\mathcal{G}_1$, i.e., it is the second highest probability graph in which the query is identifiable. Continuing in this manner, running the algorithm $n$ times would give us a ranking of the $n$ highest probability graphs.

## D Heuristic Algorithms

Algorithm 2 was devised considering the fact that every hedge formed for $Q[Y]$ must include a vertex that has a bidirected edge to $Y$. As mentioned in Section 4.2, an analogous approach, summarized in Algorithm 4, uses the fact that any hedge formed for $Q[Y]$ must include a parent of $Y$.

Let $Y \subseteq V^{\mathcal{G}}$ be a set of vertices of $\mathcal{G}$ such that $\mathcal{G}[Y]$ comprises of only one district. Let $Z := \{z \in V^{\mathcal{G}} | \exists\, y \in Y : (z,y) \in E_d^{\mathcal{G}}\} \setminus Y$ denote the set of vertices that have at least one directed edge to a vertex in $Y$, i.e., the parents of $Y$ excluding $Y$. Any hedge formed for $Q[Y]$ contains at least one vertex of $Z$. As a result, in order to eliminate all the hedges formed for $Q[Y]$, it suffices to ensure that none of the vertices in $Z$ appear in the final hedge. To this end, for any $z \in Z$, it suffices to either remove all the directed edges between $z$ and $Y$, or eliminate all the bidirected paths from $z$ to $Y$. The problem of eliminating all bidirected paths from $Z$ to $Y$ can be cast as a minimum cut problem between $Z$ and $Y$ in the edge-induced subgraph of $\mathcal{G}$ over its bidirected edges. To add the possibility of removing the directed edges between $Z$ and $Y$, we add an auxiliary vertex $z^*$ to the graph and draw a bidirected edge between $z^*$ and every $z \in Z$ with weight $w = \sum_{y \in Y} w_{(z,y)}$, i.e., the sum of the weights of all directed edges between $z$ and $Y$. Note that $z$ can have directed edges to multiple vertices in $Y$. We then solve the minimum cut problem for $z^*$ and $Y$. If an edge between $z^*$ and $z \in Z$ is in the solution to this min-cut problem, it translates to removing all the directed edges from $z$ to $Y$ in the original problem. Note that we can run the algorithm on the maximal hedge formed for $Q[Y]$ in $\mathcal{G}$ rather than $\mathcal{G}$ itself.

---

**Algorithm 4** Heuristic algorithm 2.

---

1: **function HEID2**$(\mathcal{G}, Y, W_{\mathcal{G}})$
2: $\quad$ $\mathcal{G}' \leftarrow \mathbf{MH}(\mathcal{G}, Y)$
3: $\quad$ $Z \leftarrow \{z \in V^{\mathcal{G}'} | \exists y \in Y : (z,y) \in E_d^{\mathcal{G}'}\} \setminus Y$
4: $\quad$ $\mathcal{H} \leftarrow$ The induced subgraph of $\mathcal{G}'$ on its bidirected edges.
5: $\quad$ $W_{\mathcal{H}} \leftarrow \{w_e \in W_{\mathcal{G}} | e \in \mathcal{H}\}$
6: $\quad$ $V^{\mathcal{H}} \leftarrow V^{\mathcal{H}} \cup \{y^*, z^*\}$
7: $\quad$ **for** $z \in Z$ **do**
8: $\quad\quad$ $E^{\mathcal{H}} \leftarrow E^{\mathcal{H}} \cup \{z^*, z\}$
9: $\quad\quad$ $W_{\mathcal{H}} \leftarrow W_{\mathcal{H}} \cup \{w_{\{z^*, z\}} = \sum_y w_{(z,y)}\}$
10: $\quad$ **for** $y \in Y$ **do**
11: $\quad\quad$ $E^{\mathcal{H}} \leftarrow E^{\mathcal{H}} \cup \{y, y^*\}$
12: $\quad\quad$ $W_{\mathcal{H}} \leftarrow W_{\mathcal{H}} \cup \{w_{\{y, y^*\}} = \infty\}$
13: $\quad$ $E \leftarrow MinCut(\mathcal{H}, W_{\mathcal{H}}, z^*, y^*)$
14: $\quad$ **return** $(E, \sum_{e \in E} w_e)$

---

# E Experiments

Noting that the synthetic/simulation results in the main paper were for graphs with a $\log(n)/n$ sparsity constraint, we begin this section by providing a set a results on the simulated graphs without the sparsity penalty for comparison. Then, we provide information about the causal discovery algorithm used to derive the psychology 'Psych' real-world graph. We also provide experimental results for Problem 2 formulation in Section E.3

## E.1 Additional Simulation Results without Sparsity Constraint

The simulation results for graphs generated without the sparsity constraint are shown in Figure 8. These results illustrate monotonic increases in runtime and cost as the number of nodes increases. Our proposed heuristic algorithms (HEID-1 and HEID-2) maintain runtimes less than 0.5 seconds even for 250 nodes. In contrast, the two exact algorithms (MCIP-exact and EDGEID) exceed the three minute runtime limit at only 20 nodes, and the MCIP heuristic variants (MCIP-H1 and MCIP-H2) have runtimes which increase exponentially with the number of nodes. These results highlight the efficiency of our proposed heuristic algorithms to find solutions with equivalent cost with significantly faster runtimes.

## E.2 Psychology Graph Discovery

The settings for deriving the putative structure used on the psychology real-world graph are provided in Table 3.

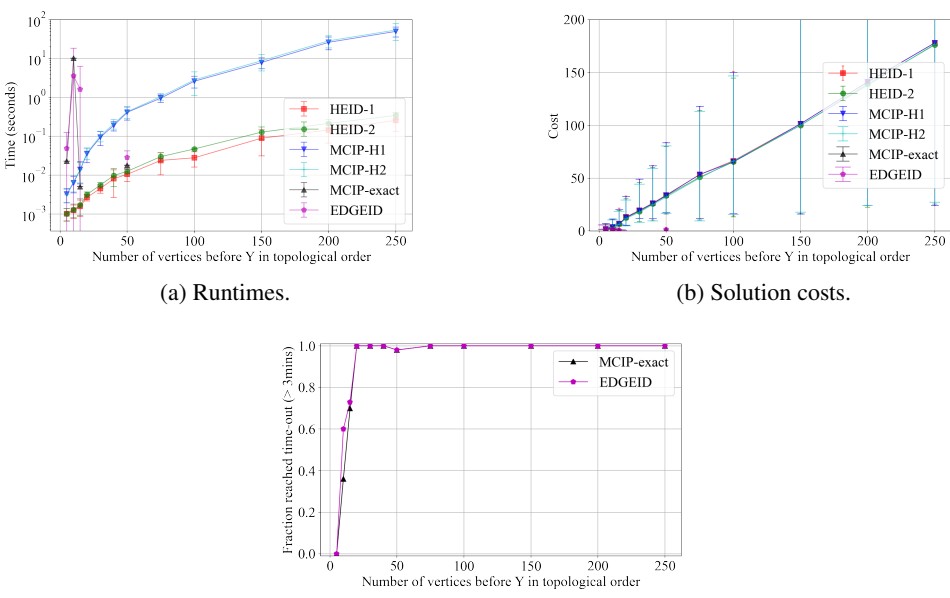

(a) Runtimes.

(b) Solution costs.

(c) Fraction for which runtime of 3 minutes exceeded.

Figure 8: Experimental results (for graphs generated without the sparsity constraint) for runtime, solution costs, fraction of graphs for which no solution was found, and fraction of graphs for which runtime limit of 3 minutes was exceeded. Error bars for runtime and cost graphs indicate 5th and 95th percentiles. Best viewed in color.

Table 3: Hyperparameter settings for the Structural Agnostic Model used to generate the putative (directed) structure for the 'Psych' real-world dataset.

| Parameter | Value |
|---|---|
| Learning Rate | 0.01 |
| DAG Penalty | True |
| DAG Penalty Weight | 0.05 |
| Number of Runs | 50 |
| Train Epochs | 3000 |
| Test Epochs | 800 |
| Mixed Data | True |
| hlayers | 2 |
| dhlayers | 2 |
| lambda1 | 10 |
| lambda2 | 0.001 |
| dlr | 0.001 |
| linear | False |
| nh | 20 |
| dnh | 200 |

### E.3 Simulation Results for Problem 2 Formulation

The experimental setup is exactly as in the main text (the results depicted in Figure 4), except that the probabilities are chosen in the range $[0.01, 1]$ instead of $[0.51, 1]$, and we use the weight mapping corresponding to Problem 2 as described in Lemma 1. That is, we map each probability $p_e$ to the weight $-\log(1 - p_e)$ in the corresponding edge ID problem.

The simulation results are presented in Figure 9. Runtimes and costs are shown for the subset of graphs for which all algorithms found a solution (to facilitate comparison). Runtimes for each algorithm are shown in Fig. 9a, where it can be seen that our proposed HEID-1 and HEID-2 heuristic algorithms have negligible runtime. In contrast, EDGEID had large runtime variance which depended heavily on the specifics of the graph under evaluation, particularly for graphs with fewer vertices.

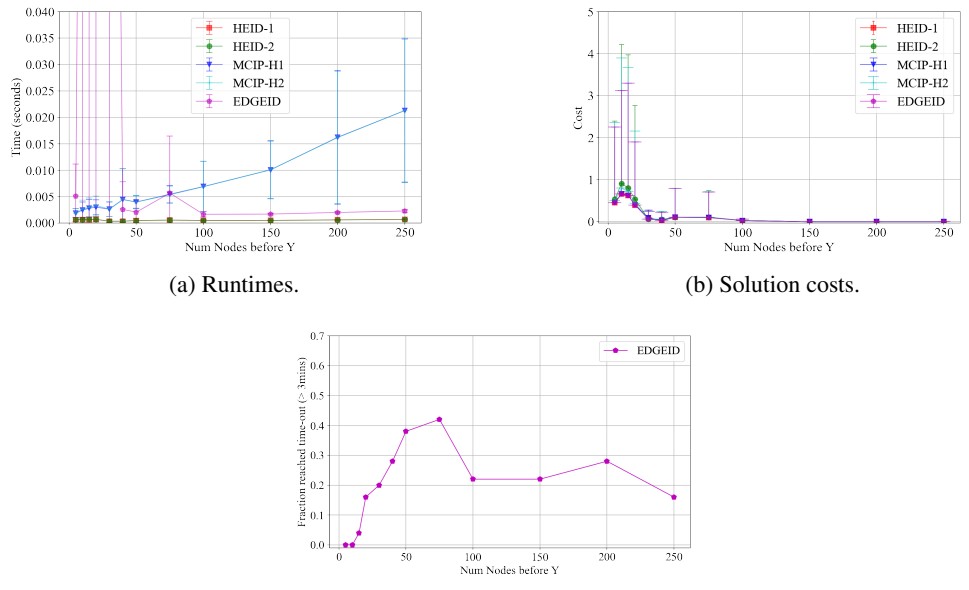

(a) Runtimes.

(b) Solution costs.

(c) Fraction runtime exceeded 3 min.

Figure 9: Experimental results for runtime, solution costs, fraction of graphs for which no solution was found, and fraction of graphs for which runtime limit of 3 minutes was exceeded. Error bars for runtime and cost graphs indicate 5th and 95th percentiles. Best viewed in color.

The costs for each graph are shown in Fig. 9b. Figure 9c shows the fraction of evaluations for which the runtime exceeded 3 minutes (applicable to the exact algorithms). In general, and owing to the sparsity penalty in our graph generation mechanism, the cost of identified solutions falls with the number of vertices. Overall, HEID-1 was both the most consistent in terms of finding a solution, having a short runtime, and achieving a close to optimal cost.

