# OpenReview forum: "Causal Discovery in Probabilistic Networks with an Identifiable Causal Effect"
_NeurIPS.cc/2022/Conference — NeurIPS 2022 Submitted_

### Official Review · Reviewer_s5gw · 2022-07-08

**Rating:** 7
**Confidence:** 4
**Soundness:** 4 excellent
**Presentation:** 3 good
**Contribution:** 3 good

**Summary:**

1. Given uncertainties about the existence of edges in a causal diagram, this paper consider two problems:

* finding the most probable graph that renders a desired causal query identifiable.

* finding the graph with the highest aggregate probability over its edge-induced subgraphs that renders a desired causal query identifiable.

2. Then, the authors introduce a combinatorial optimization problem which they call the edge ID problem and show that it is NP-hard. Further, they show that the two problems mentioned above are also NP-hard by reducing them to the edge ID problem.

3. Finally, the authors propose several exact and heuristic algorithms for the aforementioned problems.

**Questions:**

Questions:
1. In line 52, the authors say 'selecting the most plausible subgraph in which the assumption holds true'. Could they justify or elaborate on why this is the case? In other words, I fail to see the point being made here.

2. What happens when the uncertainties associated with the edges are such that $p_e = 0$ or $p_e = 1$ instead of $p_e \in [0,1]$? It is clear that the weights in Lemma 1 are either 0 or $\infty$. Would the recursive algorithm or the heuristic algorithms simplify? Would the solutions of three problems considered in this work simplify? Would the solutions/algorithms map to some existing work?

3. Could the authors provide more insight regarding Lemma 1? More specifically, is there any intuition behind the weights of the edge ID problem that Problems 1 and 2 map to?

4. Depending on the number of vertices and edges in the underlying causal diagram, the probability in (1) for any subgraph might be very small. For example, the values for Problem 1 for $G_1$ and $G_2$ are 0.049 and 0.081 respectively. Further, it is also likely that the aggregate probabilities in Problem 2 for two graphs are not very different. In that case, instead of a single identification formula, it might be useful to find a bunch of such formulae. In other words, it might be useful to derive interval bounds for the causal effect than a point estimate. Could the uncertainties about the existence of the edges and/or the problems analyzed in this work be useful towards this?

5. As mentioned in lines 148-149, Problem 2 is a surrogate to recovering the identification formula with the highest plausibility. Do the authors have a sense of how far away is this lower bound from the true value?

6. I understand that the Assumption 1 could be expanded to allow edges with perfect negative or positive correlation as mentioned in lines 158-159. But a perfect correlation between two edges basically makes the probability of one of the two edges irrelevant/useless. I don't have any major concern with Assumption 1 but it may be useful to comment on how relevant this assumption is in practice.

Suggestions / typos:
1. The authors might want to introduce certain notation before using them e.g., $P_X(Y)$ in line 47, $P_X^M(Y)$ in line 97, ...
2. The authors introduce the notation $[G]_{Id(P_X(Y))}$ in line 102 as the set of ADMGs in which $P_X(Y)$ is identifiable. Is each element of this set an edge-induced subgraph of $G$? The authors might want to clarify this.
3. In line 111, the authors mention that the set including $z$ and $t$ forms a hedge for $Q[x]$. Is this correct? Should $x$ not be a part of the hedge for $Q[x]$?
4. The authors might want to clarify 'the ID algorithm' mentioned on line 192.
5. Based on lines 268-276, it seems that there exists a hedge for $Q[Y^{mcip}]$ in ADMG $H$ in Figure 2b. Is this true? Does $Y^{mcip}$ form a district in $H[Y^{mcip}]$?
6. What is $n$ on lines 289 and 320?
7. It might be useful to provide more details about the real-world datasets used in this work. Why are the outcome variables in these datasets not fixed? Is there a particular reason why these specific datasets were used?

**Limitations:**

In its current form, this work has some limitations which are not addressed. For example, see the two points made in the weaknesses above.

Further, I am also concerned about the relevance of the Problems studied here. In Problem 1, the authors are interested in finding the most probable causal diagram in which the causal effect is identifiable. In Problem 2, the authors are interested in finding (a lower bond on) the most probable causal effect. These problems makes sense when there exists no way to find the causal effect (under the assumption that only the uncertainties associated with the edges are known).

While most algorithms proposed to identify causal effect assume access to a correctly specified causal diagram, there has been a line of work for identifying causal effect using only an anchor variable when the underlying causal diagram is not known. For example: (1) Entner et al 2013 -- Data-driven covariate selection for nonparametric estimation of causal effects [https://proceedings.mlr.press/v31/entner13a.html], (2) Cheng et al 2020 -- Towards unique and unbiased causal effect estimation from data with hidden variable [https://arxiv.org/pdf/2002.10091.pdf], (3) Shah et al 2021 -- Finding Valid Adjustments under Non-ignorability with Minimal DAG Knowledge [https://arxiv.org/pdf/2106.11560.pdf].

In light of these works, I am concerned about the relevance of the two problems studied in this work. First, by requiring uncertainty for *every* edge in a causal diagram, this paper require global knowledge of the underlying causal diagram which may be too much to ask for as described in first weakness above. In contrast, the aforementioned works require only local knowledge of the underlying causal diagram concerning the anchor variable. Second, and more importantly, consider a scenario where the aforementioned works are able to find a valid identification formula for a causal effect of interest. In that case, no matter what the rest of the causal diagram looks like, the identification formula remains valid. On the other hand, this may not be true for the two problems studied in this work. Consider the example 1 on page 4. If one were to solve Problem 1, they may choose $F_2$ over $F_1$ and if one were to solve Problem 2, they may choose $F_1$ over $F_2$. However, it is certainly possible that the true underlying causal diagram is $G$ in Figure 1a and neither $F_1$ nor $F_2$ are valid identification formulae. To conclude, what is the validity of the causal effect / identification formula returned by the two problems considered in this work? I understand that this may not be apples to apples comparison with the three works mentioned above, my question in the previous sentence stands. How would the outputs of the Problems considered here be used in a real-world application?

**Strengths And Weaknesses:**

Strengths:
1. This paper introduces three new problems and prove that they are NP-hard:
* finding the most probable graph that renders a desired causal query identifiable.
* finding the graph with the highest aggregate probability over its edge-induced subgraphs that renders a desired causal query identifiable.
* finding the set of edges $E$ with minimum aggregated weight that renders a desired causal query identifiable in the graph resulting from removing $E$ (i.e., the edge ID problem).
2. As a consequence of their work, the authors also show that if an identification formula is valid w.r.t a causal graph, it is also valid w.r.t all its edge-induced subgraphs.
3. Further, they also establish a bi-directional link between the edge ID problem and the minimum-cost intervention problem. Therefore, the algorithms proposed here could be useful to solve the minimum-cost intervention problem as well.

Originality:

The paper is technically solid and brings a significant originality of ideas. However, I am a little concerned about the importance of the questions considered. See the limitation section below.

Clarity:

The paper is mostly well-written. There are certain places where the writing could be improved/made clear. Please see 'Suggestions / typos' below.

Weakness:
1. **Relevance of the problem setup:** As mentioned in the paper, most algorithms proposed to identify causal effect assume access to a correctly specified causal diagram. This assumption includes knowing the presence of every edge in a causal diagram with probability 0 or with probability 1 i.e., $p_e = 0$ or $p_e = 1$. While this paper does not need to know a binary $p_e$, it still needs to know a continuous valued $p_e$ i.e., $p_e \in [0,1]$ for every edge in a causal diagram. In my opinion, such information is difficult to obtain in most real-world applications. The authors sample $p_e$ uniformly between 0.51 and 1.0 in all of their experiments and mention that these uncertainties may reflect the confidence of a particular statistical test. However, how would one obtain such information for bi-directed edges even from statistical tests? I would encourage the authors to think more carefully about this requirement and include real-world scenarios where this requirement would be satisfied.

2. **Type of causal queries studied:**  In Remark 1, the authors mention that their results are not limited to causal queries of the form $Q[Y]$. They can be applied to general queries of the form $P_X(Y)$ when the directed edges of the causal diagram are known. However, in this case, the authors are inherently eliminating any uncertainties associated with the directed edges of the causal diagram which seems against the spirit of the paper.

---

> ### Author Response · Authors · 2022-08-01
> **Response to Reviewer s5gw**
>
> We thank the reviewer for their thorough feedback. We appreciate the reviewer finding our work solid and original.
> We would like to first address the main concerns raised by the reviewer:
> ---
> >  How would one obtain such information for bi-directed edges even from statistical tests?
>
> We appreciate the reviewer's concern about obtaining edge probabilities.
> However, we would like to mention that the edge probabilities over bidirected edges can also be the result of both statistical tests, and additional expert knowledge.
> This can for instance happen when we have access to interventional data, rather than mere observational study.
> Indeed, learning an ADMG does require interventional data (otherwise it is learnable only up to Markov equivalence.)
> To give a concrete example, let us assume that we want to tell the difference between the two graphs $x\to y$ and $x\leftrightarrow y$.
> Comparing the confidence of the independence tests before and after intervening on $x$ can provide us a probability on both of the edges $x\to y$ and $x\leftrightarrow y$.
>
> >While this paper does not need to know a binary $p_e$, it still needs to know a continuous valued $p_e$ i.e., $p_e\in[0,1]$ for every edge in a causal diagram. In my opinion, such information is difficult to obtain in most real-world applications.
>
> As we shall discuss in limitations below, slight variations of our method are applicable even in the case that merely local information is available.
>
> > The authors mention that their results are not limited to causal queries of the form
> $Q[Y]$. However, the authors are inherently eliminating any uncertainties associated with the directed edges of the causal diagram.
>
> We agree with the reviewer that knowing the ancestors of $Y$ is restrictive.
> However, we would like to point out that the assumption that the ancestors of $Y$ are known in $\mathcal{G}\setminus X$ is not as restrictive as eliminating 'any uncertainty' associated with the directed edges.
> This assumption is in fact in the spirit of perfect positive/negative correlation constraints discussed above Assumption 1.
> We will elaborate more by an example.
> Consider the following graphs:
> 1) $z\to y\gets t$,
> 2) $z\to t\to y$,
> 3) $z\to t\to y$, $z\to y$.
>
> All three of them share the same ancestral information, i.e., that both $z,t$ are ancestors of $y$.
> As shown in this example, knowing the ancestors of $y$ does not preclude all the uncertainty on the directed edges (as fixing the edge probabilities to 0 and 1 does), but it rather imposes a constraint of the type that if the edge $z\to y$ does not exist, the path $z\to t\to y$ must.
> We have further elaborated this in the revision, through Fig. 2, and lines 166-170.
> ---
> limitations:
>
> We thank the reviewer for bringing up this interesting line of research.
> We agree with the reviewer that the comparison between those works and ours is not apple to apple.
> A significant difference is that they look only for a certain type of identification formula, namely an adjustment set.
> For instance in the following graph, there is no valid adjustment set for $P_x(y)$, although it is identifiable:
> $w\to m\to x\to y$, $x\leftrightarrow w\leftrightarrow y$.
> This is the reason that local information is not enough to derive all possible types of identification formulae.
> Another important issue to mention is that some of the work mentioned by the reviewer require additional assumptions (such as knowing at least one parent of the outcome variable as a priori), whereas our approach does not require such assumptions.
> It is also worth mentioning that the adjustment sets found by the aforementioned work are based on statistical tests (e.g., conditional independence tests) which can be erroneous due to lack of data.
> This was indeed one of the motivations that we proposed modeling edge probabilities based on the confidence of statistical tests in our work.
> Further, even if merely local information about the graph is accessible, slight variations of our approach are still applicable.
> For instance, if we have no information about the existence of a bidirected edge $e$, we can assign probability 1 to its existence and carry out our algorithms.
> If our algorithms return a graph $\mathcal{G}$ in which the causal query is identifiable, $\mathcal{G}$ will contain $e$, and this is in fact to say that the query is identifiable regardless of the existence of $e$.
> (due to proposition 1.)
>
> On a separate note, our approach is more flexible when it comes to interventional studies.
> For instance, consider the setup described in [3]: we would like to find a set of variables to intervene upon so that a certain query becomes identifiable.
> None of the work mentioned by the reviewer can provide information about this question before actually performing the intervention, whereas our approach could be utilised to answer such questions before any experiment.
>
> [3]  S. Akbari, J. Etesami, and N. Kiyavash. Minimum cost intervention design for causal effect identification.

---

> > ### Author Response · Authors · 2022-08-01
> > **Response to Reviewer s5gw continued**
> >
> > We would also like to address the questions asked by the reviewer:
> >
> > ---
> > > In line 52, the authors say 'selecting the most plausible subgraph in which the assumption holds true'. Could they justify or elaborate on why this is the case? In other words, I fail to see the point being made here.
> >
> > The point we were trying to make here is that it is common in some applications to assume ignorability in order to perform causal inference.
> > We propose making a weaker assumption which still allows us to do inference.
> > In fact, our setup quantifies the strength of any structural assumption (including non-confoundedness), and one can choose the weakest assumption for the application of their interest.
> > Moreover, this provides the researcher with a quantitative measure of how strong the assumption they are making for their following arguments would be.
> > We have rephrased this sentence to be clearer.
> >
> > >What happens when the uncertainties associated with the edges are such that $p_e=0$ or $p_e=1$ instead of $p_e\in[0,1]$? Would the recursive algorithm or the heuristic algorithms simplify?
> >
> > In the case that all of the probabilities are either 0 or 1, then the only subgraph of $\mathcal{G}$ with non-zero probability is $\mathcal{G}$ itself.
> > Indeed all of the problems considered in this work then map to the problem of whether the causal query is identifiable in $\mathcal{G}$ or not, which is the classic identifiability problem considered in the literature (for instance, [2]).
> >
> > [2] Complete identification methods for the causal hierarchy, Shpitser and Pearl, JMLR2008.
> >
> > > Could the authors provide more insight regarding Lemma 1? More specifically, is there any intuition behind the weights of the edge ID problem that Problems 1 and 2 map to?
> >
> > The weights in Lemma 1 are chosen in a way to transform the probability maximisation problem into a problem with additive costs.
> > The logarithmic transforms serve to this matter.
> > We have included a brief explanation of this point in our revised version (above lemma 1).
> >
> > > It might be useful to find a bunch of such formulae. In other words, it might be useful to derive interval bounds for the causal effect than a point estimate. Could the uncertainties about the existence of the edges and/or the problems analyzed in this work be useful towards this?
> >
> > We thank the reviewer for this interesting question.
> > The approach discussed in this paper can indeed be used towards solving such a problem.
> > Although this can be considered as a future direction, a naive brute-force approach would be as follows.
> > Let us assume we run our algorithm which returns the subgraph with the highest probability, $\mathcal{G}_1$.
> > Based on what we proposed in Appendix C, perfect negative correlation constraints can be added to the edge ID problem without additional computational cost.
> > So we solve the original problem for a second time (i.e., re-run the algorithm), with the only difference that we add the perfect negative correlation constraint over the set of all edges of $\mathcal{G}_1$ (i.e., we force the algorithm to exclude at least one of the edges of $\mathcal{G}_1$.)
> > The solution to this problem (let us call it $\mathcal{G}_2$) is the highest probability graph among all subgraphs except $\mathcal{G}_1$, i.e., it is the second highest probability graph.
> > Continuing in this manner, running the algorithm $n$ times would give us a ranking of the $n$ highest probability graphs.
> > A combination of these graphs would then be a solution to the problem that the reviewer suggests.
> > We have included this explanation in Appendix C.1 of our revision.
> >
> > > Problem 2 is a surrogate. Do the authors have a sense of how far away is this lower bound from the true value?
> >
> > Aside from the computational considerations, there is a theoretical barrier to evaluate such a distance:
> > the existing identification algorithms in the literature provide only one identification formula per graph and query, and not all such formulae.
> > We do not have a clear answer for this question.
> > As a result, we suggest relying on results such as proposition 1 to approach problem 2.
> >
> > >A perfect correlation between two edges basically makes the probability of one of the two edges irrelevant/useless. I don't have any major concern with Assumption 1 but it may be useful to comment on how relevant this assumption is in practice.
> >
> > We agree that this expansion is not very general, but as mentioned in our response to the question above, it can be helpful to formulate further problems in terms of the problems considered in this work.
> > We refer to Appendix C.1 in our revised paper.
> > ---
> > We also acknowledge the reviewer's detailed suggestions. We have applied these suggestions and fixed the typos in the revision:
> > - missing notation is introduced,
> > - definition of $[\mathcal{G}]_{Id(P_X(Y))}$ is made clearer,
> > - the typo in line 111 is fixed,
> > - ID algorithm is explicitly clarified,
> > - further explanation on $Y^{mcip}$ is included, and the parameter $n$ is clarified.

---

> > ### Comment · Reviewer_s5gw · 2022-08-08
> > **Response to author rebuttal**
> >
> > I would like to thank the authors for their detailed rebuttal. The authors answer most of my questions/concerns and I would like to increase my score.
> >
> > - Re edge probabilities over bidirected edges: I can see from the example how such edge probabilities could potentially be obtained with interventional data. However, I am not sure how practical/feasible this approach is. That being said, I understand that not everything needs to be addressed straight away. I would encourage the authors to add their response to the revised version.
> >
> > - Re eliminating any uncertainties associated with the directed edges: Thanks for the example and for correcting me!
> >
> > - Re comparison with the line of work mentioned in the original review: I am happy with the author response. The authors identify scenarios where their work is relevant/applicable and the works mentioned in the original review are not. Please add this discussion in the revised version.
> >
> > - Re responses to the questions in the original review: Thanks for all the satisfactory answers! Please add the discussion regarding 'a weaker assumption compared to ignorability' in the revision (if not already).

---

### Official Review · Reviewer_Yekr · 2022-07-10

**Rating:** 5
**Confidence:** 4
**Soundness:** 3 good
**Presentation:** 4 excellent
**Contribution:** 2 fair

**Summary:**

The paper considers a setting where one does not have access to a fully specified causal ADMG but a probabilistic causal graph in which the directed/bidirected edges are assigned probabilities. It studies two questions under this setting:  Given a probabilistic graph and a specific causal effect of interest, (1)  which subgraph has the highest plausibility and (2) which subgraph has the highest aggregate probability over its edge-induced subgraphs for which the causal effect is identifiable? The paper shows that both problems reduce to an NP-hard combinatorial optimization problem, provides several exact and heuristic algorithms, and empirically evaluates the proposed algorithms.


**Questions:**

Are the results limited to causal queries of the form Q[Y] or applicable to arbitrary causal effects $P_X(Y)$? Please clarify Remark 1.

In the experiments, the edge probabilities $p_e$ are sampled uniformly between 0.51 and 1.0. Why making $p_e > 0.5$? I believe this makes the graph samples biased. In fact, I suspect making all edge probabilities larger than 0.5 might make certain optimization problems easy to solve (just a guess), but any reason keeping $p_e > 0.5$?

Line 111, {z, t} should be {z, x}?


**Limitations:**

If the results are indeed limited to causal queries of the form Q[Y] instead of arbitrary causal effects $P_X(Y)$, then this limitation should be made clear in the abstract/introduction, etc.


**Strengths And Weaknesses:**

Originality

The paper introduces a new causal inference setting represented by probabilistic causal graphs that capture uncertainties in the existence of edges. The setting relaxes the restrictive requirement that one must have access to a fully specified causal graph in the majority of works on causal effect identification. To the best of my knowledge, this problem setting is novel.

Quality of work

The technical results look sound. The proposed algorithms look reasonable. The empirical evaluation is largely adequate.

Clarity

The paper is clearly written. The presentation is excellent.

Significance

The main weakness of the paper is in its significance. Although the results are of some interest, it looks to me the paper does not really solve the problem it claims to be solving.

-It looks to me that the results are limited to causal queries of the form Q[Y] instead of arbitrary causal effects $P_X(Y)$. Remark 1 appears to address this issue and claims ``They can be applied to general causal queries of the form $P_X(Y )$ if the directed edges of G and therefore the set $Anc_{G\setminus X}(Y)$ are known.'' It’s not clear what this claim means. Are the results of the paper applicable to general $P_X(Y )$ only if all the directed edges are fixed with probability 1?

-Problem 2 does not solve the problem of deriving an identification formula for a given causal query that holds with the highest probability. Solving Problem 2 finds a lower bound on the plausibility of an identification formula since there may exist other types of graphs with the same identification formula.

---

> ### Author Response · Authors · 2022-08-01
> **Response to Reviewer Yekr**
>
> We thank the reviewer for their time and constructive feedback.
> We would like to address the main concerns raised by the reviewer:
>
> ---
> >The main weakness of the paper is in its significance. Although the results are of some interest, it looks to me the paper does not really solve the problem it claims to be solving.
>
> >Problem 2 does not solve the problem of deriving an identification formula for a given causal query that holds with the highest probability. Solving Problem 2 finds a lower bound on the plausibility of an identification formula since there may exist other types of graphs with the same identification formula.
>
> The surrogate problem considered instead of original problem 2 maximises a lower bound of problem 2.
> Although there is no guarantee that this lower bound is close to the real value, if the corresponding probability is high enough, it suffices for practical purposes.
> We would like to note that for problem 2, it is crucial to have an understanding of all the valid identification formulae for a given causal query with respect to a graph.
> To the best of our knowledge, this has not been studied yet, and is an interesting open problem (Note that the space of all formulae is not tractable.)
> We have discussed this further (lines 138 through 146) of our revised version.
> In this work, we proposed a tractable, surrogate problem, where its solution is a lower bound to the original problem.
>
> On a separate note, we have included a method to provide a ranking of the (for instance, 10 most probable graphs) based on the suggestion of Reviewer s5gw in Appendix C.1 of our revised paper.
> This could also serve as yet another surrogate to solve the problem of finding the most probable formula.
>
> > It looks to me that the results are limited to causal queries of the form $Q[Y]$ instead of arbitrary causal effects $P_X(Y)$. Remark 1 appears to address this issue and claims ``They can be applied to general causal queries of the form $P_X(Y)$ if the set $Anc_{\mathcal{G}\setminus X}(Y)$ is known.'' It’s not clear what this claim means. Are the results of the paper applicable to general $P_X(Y)$ only if all the directed edges are fixed with probability 1?
>
>
> The assumption that the ancestors of $Y$ are known in $\mathcal{G}\setminus X$ is not as restrictive as fixing the probability of all the directed edges to 1.
> This assumption is in fact in the spirit of perfect positive/negative correlation constraints discussed above Assumption 1.
> We will elaborate more by an example.
> Consider the following graphs:
> 1) $z\to y\gets t$,
> 2) $z\to t\to y$,
> 3) $z\to t\to y$, $z\to y$.
>
> All three of these graphs share the same ancestral information, i.e., that both $z$ and $t$ are ancestors of $y$.
> As shown in this example, knowing the ancestors of $y$ is not equivalent to precluding uncertainty on the directed edges (as in the case of fixing the edge probabilities to 0 and 1), but it rather imposes a constraint of the type that if the edge $z\to y$ does not exist, the path $z\to t\to y$ must.
> We have further elaborated this in the revision, through Fig. 2, and lines 166-170.
>
> ---
> Questions:
>
> >In the experiments, the edge probabilities $p_e$ are sampled uniformly between 0.51 and 1.0. Why making
> $p_e>0.5$? I believe this makes the graph samples biased. In fact, I suspect making all edge probabilities larger than 0.5 might make certain optimization problems easy to solve (just a guess), but any reason keeping $p_e>0.5$?
>
> The choice of $p>0.5$ in our experiments is due to Lemma 1, as any edge with probability less than 0.5 would be mapped into an edge with zero weight in the equivalent edge ID problem, which can always be removed at the beginning of the algorithm. We clarified this point in our revised version (footnote 3 on page 8).
>
> > Line 111, {z, t} should be {z, x}?
>
> We thank the reviewer for pointing this typo out. We have fixed it in our revised version.

---

> > ### Comment · Reviewer_Yekr · 2022-08-05
> > **question**
> >
> > ``The choice of $p>0.5$ in our experiments is due to Lemma 1, as any edge with probability less than 0.5 would be mapped into an edge with zero weight in the equivalent edge ID problem, which can always be removed at the beginning of the algorithm.''
> >
> > It looks to me, by Lemma 1, this claim only holds for Problem 1, not for Problem 2.

---

> > > ### Author Response · Authors · 2022-08-08
> > > **Response**
> > >
> > > > It looks to me, by Lemma 1, this claim only holds for Problem 1, not for Problem 2.
> > >
> > > Exactly. Indeed, we had provided simulation results for the formulation of Problem 1 for demonstrative purposes, as both Problems 1 and 2 are reduced to the edge ID problem, only with different weight mappings.
> > > For the sake of completeness, we have added simulation results for Problem 2 formulation in Appendix E.3.

---

### Official Review · Reviewer_PiUW · 2022-07-10

**Rating:** 6
**Confidence:** 4
**Soundness:** 3 good
**Presentation:** 3 good
**Contribution:** 3 good

**Summary:**

This paper addresses the problem of causal identification under structural uncertainty. In particular the authors assume that the probability of edge existence is given and provide an algorithm to find the most probable configuration for a given causal query of interest. Empirical results show promise.

**Questions:**

In many practical settings I would imagine that there are a large number of dense graphs that appears (i.e., many non-zero probabilities) do the authors have ideas or guidance on how to reduce the size of the search space?

**Limitations:**

The authors address limitations of the proposed approach well.

**Strengths And Weaknesses:**

Strengths:

* Novel and interesting task. Structural uncertainty and misspecification is commonplace, and addressing this for identification is a very important problem area.
* Well described problem setup and the authors provide both exact and heuristic solutions.

Weaknesses:

* Approach seems fraught in dense graphs.
* Experimental evidence could be more compelling, would especially like to see a comparison with more naive approaches.

---

> ### Author Response · Authors · 2022-08-01
> **Response to Reviewer PiUW**
>
> We thank the reviewer for their time and feedback. We appreciate the reviewer finding our work novel and interesting.
>
> ---
> Regarding the question of the reviewer concerning dense graphs,
> > In many practical settings I would imagine that there are a large number of dense graphs that appears (i.e., many non-zero probabilities) do the authors have ideas or guidance on how to reduce the size of the search space?
>
> Since we proved the NP-hardness of the edge ID problem, we would expect the computational cost of any exact approach to solve these problems to be high.
> Therefore, we proposed two heuristic algorithms for approximating the solutions.
> These algorithms are polynomial-time and run exceptionally fast in comparison with the exact approaches.
> This performance advantage is shown in Figure 7, which shows the performance of our algorithms on large and dense graphs.

---

### Official Review · Reviewer_2dMf · 2022-07-11

**Rating:** 6
**Confidence:** 4
**Soundness:** 4 excellent
**Presentation:** 3 good
**Contribution:** 2 fair

**Summary:**

This paper addresses problems of probabilistic reasoning about uncertainty in causal diagrams. In particular, given a simple distribution over ADMGs, and a causal effect query, the authors consider 1) finding the most likely graph that is identifiable; and 2) finding the identifiability formula with the greatest probability of being valid. They show that 1) and a relaxation of 2) are equivalent to the edge-ID problem, which they show to be NP-hard (in particular, equivalent to the minimum-cost intervention problem). In view of this, they provide a number of heuristic algorithms to solve the edge-ID problem, demonstrating their efficacy on both simulated graphs and real-world datasets.

**Questions:**

1) What practical applications can the authors envisage for their methods?
2) In all experiments, the probability of an edge is restricted to lie in [0.51, 1]. Is this because an edge with probability <0.5 would always be removed? If so, do the authors consider this a limitation of their formulation of the problem (in that positive correlations between edges are not possible)?

**Limitations:**

The authors appropriately acknowledge the limitations of their method (independent edges, approximation made by Problem 2).

**Strengths And Weaknesses:**

Overall, this is a technically strong and interesting paper which introduces a number of new ideas. The authors introduce the novel problem of evaluating (maximal) probabilities of identifiability under graph uncertainty, and provide both clear formal results on hardness as well as practical heuristic algorithms.

Form a theoretical perspective, the reductions to and from MCIP (Thm 1 and Thm 2) are quite non-trivial and provide an interesting computational link between the problems of choosing minimal cost interventions and most likely edge-subgraphs for identifiability. The authors study the problem under the restrictive assumption of independent edges, but also introduce a clever extension to distributions with perfect negative correlations. Algorithmically, the backtracking exact algorithm is simple but sensible, while the heuristic algorithms make clever use of the properties of hedge structures.

The presentation of the paper has clearly been thought out carefully; in particular, the technical clarity of the main paper and appendix proofs is excellent. However, in terms of telling the story, the paper as written seems to rely on a lot of implicit context to be fully understood, which could perhaps be spelled out more explicitly. To give a few examples, with the aim of providing constructive suggestions for improving readability:
- Definition 4 of a hedge appears to be non-standard, as a specific case of the full definition where G[Y] forms a c-component and the intervention is on all other nodes V\Y; while I understand the authors’ desire to set up the intuition, it would be helpful to explicitly state this.
- Theorem 1 appears before the MCIP problem is defined or even mentioned; in my opinion, given that the relationship between these two problems is so close as shown in the proofs of Thm 1 and 2, it would be useful to explicitly mention this and give some intuition in the main paper.
- The MCIP problem formulation is not intuitive (“minimum cost set of interventions”) without the context in [1] about why the stated formulation is consistent; this could be stated explicitly.

My main criticism of the paper is with regards to the motivation of the edge-ID problem, and as a result the degree to which the paper breaks new ground beyond [1]. In particular, it is not clear why one would want to find the most likely identifiable graph. Unless this probability is very high, it would be unreasonable to rely on identification formulae based on that graph. Finding the identification formula that is most likely to be valid is more compelling, but unfortunately (and understandably from a computational perspective) a rather crude relaxation is used, which could have very different solutions.

In this sense, I would have liked to have seen some experiments demonstrating the utility of the edge-ID problem, by putting it in the context of the original problems. In particular, it might be useful to see the raw probabilities associated with the most likely graph/most likely aggregate graphs; or perhaps, in cases where exact computation is feasible, the gap between the probability of the most likely identification formula and the solution to Problem 2.

Nonetheless, this is an extremely solid work overall that provides new understanding of the computational issues regarding probabilistic reasoning about identifiability.

[1] S. Akbari, J. Etesami, and N. Kiyavash. Minimum cost intervention design for causal effect identification.

---

> ### Author Response · Authors · 2022-08-01
> **Response to Reviewer 2dMf**
>
> We thank the reviewer for their time and feedback. We appreciate the reviewer finding our work solid.
>
> We would like to mention two points regarding the motivation of our problem:
>
> 1) Finding the most probable graph provides the most likely graph in which the causal query is identifiable, but more importantly, this is done for the weakest structural assumption under which the query is identifiable.
> Our main goal in this work is to define a quantitative measure of the strength of the structural assumption that we make prior to the inference task.
> We agree with the reviewer that if this probability is not high enough, then there is no point in using the corresponding identification formula.
> This probability gives the researcher an idea of how strong of an assumption they are making before undertaking any further inference.
> Moreover, we have added a brief application of our methods in Appendix C.1, where we show how to find a ranking of (for instance 10 most probable graphs) rather than merely on graph, which could be useful to use a combination of them (for instance, as another surrogate to problem 2).
>
> We would also like to mention the line of research by Pearl and others on testable implications of the instrumental variable model (for instance, [1])
> In these work, the proposed causal structure (namely, the IV model) is shown to be wrong based on evidence from the observed data, which in turn shows that inference based on this model is not valid.
> In contrast, we provide the probability that a valid identification can be derived, using the edge probabilities based on evidence from observed data and the expert knowledge.
>
> 2) The surrogate problem considered instead of original problem 2 maximises a lower bound of problem 2.
> Even though there is no guarantee that this lower bound is close to the real value, if the corresponding probability is high enough, it suffices for practical purposes.
> We would like to note that our choice of solving a surrogate problem was not merely due to computational considerations, but also due to the fact that to the best of our knowledge, there is no existing approach that can find `all' identification formulae corresponding to a given graph and query. We have added this explanation in our revised version, lines 138-146.
>
> [1] On the Testability of Causal Models with Latent and Instrumental Variables, Pearl, UAI'95.
>
>
> ---------------------------------------
>
> > In all experiments, the probability of an edge is restricted to lie in [0.51, 1]. Is this because an edge with probability <0.5 would always be removed? If so, do the authors consider this a limitation of their formulation of the problem (in that positive correlations between edges are not possible)?
>
> Indeed, due to Lemma 1, any edge with probability less than 0.5 would be mapped into an edge with zero weight in the equivalent edge ID problem, which can always be removed at the beginning of the algorithm.
> We agree that Lemma 1 follows from the independence assumption.
> We have clarified this point in the revised version, footnote 3 on page 8.
>
> ---------------------------------------
>
> We also acknowledge the reviewer's comments regarding 'implicit context'.
> In this regard, we made the definition of MCIP clearer, and added the statement explaining the standard definition of hedge.

---

> > ### Comment · Reviewer_2dMf · 2022-08-07
> > **Thank you for the response**
> >
> > I would like to thank the authors for their response to my concerns and comments, and for the updated manuscript. I remain positive on the paper due to the novelty of the problem and sound theoretical work, though still with some concerns on the significance in the context of realistic scenarios.
> >
> > As also highlighted by other reviewers, I would encourage the authors to conduct experiments emulating realistic scenarios in which this method could be applied, such as those which have been suggested here. In the response, the authors have suggested that the probability can be used to evaluate whether the formula should be trusted, or, as reviewer s5gw suggested, obtaining bounds on the causal effect. However, the fact that we choose the maximal probability identifiable subgraph (or aggregation), is likely to introduce a bias. That is, while the probabilistic guarantee of correctness of the formula is valid, it may be the case that when the formula is incorrect, the difference between the causal effect obtained from the chosen formula from a correct formula may be larger than if a random formula was chosen. Thus, looking at probabilities alone may not be sufficient in practice.

---

### Meta-Review · Area_Chair_vZnn · 2022-08-26

**Recommendation:** Reject
**Confidence:** Less certain

**Metareview:**

This paper studies the problem of causal identifiability in probabilistic causal models, where each edge is associated with a probability value that indicate the uncertainty about the existence of edge and whether a given causal effect is identifiable. Two technical problems are considered: 1) finding the most probable graph that renders a desired causal query identifiable, and 2) finding the graph with the highest aggregate probability over its edge-induced subgraphs that renders a desired causal query identifiable.

A reasonable amount of discussions took place between the authors and the reviewers, and among the reviewers themselves. At the end, we get four confident (4) reviews with ratings 5, 6, 6, and 7:
The reviewers appreciate the novel problem setting, interesting complexity results, reasonable algorithms and clear presentation.

However, there is also concern that since the paper solves a surrogate problem than the one it sets out to solve, there needs to be more acknowledgement and up front discussion about this gap in the paper, and discussion on potential ways to bridge this gap. In general, a broader discussion on the practical utility of the work developed here, rather than just the ideal question is expected in the final revision, along with other reviewer feedback.


**Award:**

No

---

### Decision · Program_Chairs · 2022-09-14

Reject